# Buildings hazard maps with differentiated exceedance probability for flood impact assessment

Punit K. Bhola[1], Jorge Leandro[1], Markus Disse[1]

[1]Chair of Hydrology and River Basin Management, Department of Civil, Geo and Environmental Engineering, Technical University of Munich, Arcisstrasse 21 80333 Munich, Germany

*Correspondence to*: Punit K. Bhola (punit.bhola@tum.de)

**Abstract.** In operational flood risk management, a single *best-model* is used to assess the impact of flooding, which might misrepresent uncertainties in the modelling process. We have used quantified uncertainties in flood forecasting to generate flood hazards maps that were combined based on different exceedance probability scenarios. The purpose is to differentiate the impact of flooding depending on the building use enabling, therefore, more flexibility for stakeholder's variable risk perception profiles. The aim of the study is thus to develop a novel methodology that uses a multi-model combination of flood forecasting models to generate flood hazard maps with differentiated exceedance probability. These maps take into account uncertainties steaming from the rainfall-runoff generation process and could be used by decision-makers for a variety of purposes in which the building use plays a significant role, e.g. flood impact assessment, spatial planning, early warning and emergency planning.

## 1 Introduction

Floods are one of the most destructive natural hazards and lead to severe social and economic impacts (European Union, 2007; Alfieri et al., 2016). The number of people exposed to recent flooding occurred in many Central European countries highlights the importance of assessing flood hazards. During the extensive June 2013 floods in Germany, for example, more than 80,000 people in eight federal states had to be evacuated (Thieken et al., 2016). The vulnerability of settlements calls for an improved flood forecasting, which includes underlying uncertainties and impacts.

In this study, we present a novel methodology that uses a multi-model combination of two-dimensional (2D) hydrodynamic (HD) models to assess the impact of flooding based on water depth hazards. These hazards can be evaluated for key urban features, such as buildings, roads, bridges and green spaces (Leandro et al., 2016). This study focusses in particular on buildings. Furthermore, the hazard maps serve a variety of purposes, e.g. flood impact assessment, spatial planning, early warning and emergency planning (Hammond et al., 2013) for target users. For this paper, the users consist of a group of decision-makers, such as the Bavarian Water Authorities and disaster relief organizations in Germany, the Federal Agency for Technical Relief or the German Red Cross.

In deterministic flood forecasting, the predictions of forecasting models, precipitation forecasts, hydrological models and HD models, are used to generate flood hazard maps. These maps form the basis of flood risk management and are utilised to assess the impact of floods (Schanze, 2006; Hagemeier-Klose and Wagner, 2009). Although advances are continually being made in real-time forecasting, they are still inherently uncertain (Meyer et al., 2009; Bates et. al., 2014; Beven et al., 2018). The

decision-making process based on uncertain predictions can have a huge economic impact and possibly lead to life and death situations (Leedal et al., 2010). Thus, a thorough assessment is required of the extent to which uncertainties affects the predictions of flood hazards. In addition, forecasting predictions that inform policy or risk management decisions should include major sources of uncertainty and communicate them coherently (Todini, 2017).

Researchers have addressed various sources of uncertainties in flood modelling, such as precipitation measurements, spatial

interpolation of the precipitation, model parameter, model structure (Nester et al., 2012; Leandro et al., 2013), discharge data, measured discharge and uncertainty estimation techniques (Dotto et al., 2012). Although uncertainties arising from precipitation and HD models are significant, the generation of discharges using a hydrological model is considered as one of the most uncertain steps in flood forecasting (Di Baldassarre and Montanari, 2009). Substantial research has been dedicated to the field of discharge forecasting and reducing uncertainties by using methods, such as Generalized Likelihood Uncertainty

Estimation (Beven and Binley, 2014), Global Sensitivity Analyses (Pappenberger et al., 2008) and the Shuffled Complex Evolution Metropolis Algorithm (Dotto et al., 2012). To find the appropriate method, Pappenberger et al. (2006) have provided a decision tree that helps users select a suitable method for a given solution. Furthermore, in a recent study Boelee et al. (2018) reviewed uncertainty quantification methods to provide practitioners with an overview of ensemble modelling techniques. An overview of existing ensemble forecasts in operational use can be found in Cloke and Pappenberger (2009) and Todini (2017).

Most notably, in the federal states of Rhineland-Palatinate (Bartels et al., 2017) and Bavaria (Laurent et al., 2010) discharge ensembles are generated using the COSMO-DE-EPS precipitation ensemble as input to a distributed hydrological model LARSIM (Large Area Runoff Simulation Model). These and similar developments offer a potential framework for quantifying uncertainties. A challenging issue in natural hazards, however, remains the effective communication of the quantified uncertainties to decision-makers (Doyle et al., 2019). Researchers have questioned how uncertainties should be communicated

to reduce the risk of wrong or inappropriate decisions (Bruen et al., 2010; Todini, 2017).

In operational flood forecasting, hazard maps are provided in the form of exceedance probability scenarios and generally, only one scenario is considered for emergency planning. Normally, a 50% exceedance probability scenario (or median) is expected to be close to the deterministic *best-model* approach (Di Baldassarre et al., 2010). In other examples (Beven et al., 2014; Beven et al., 2015; Disse et al., 2018), model results of various exceedance probabilities are provided on separate or combined maps.

Kolen et al. (2019) stated that there is a need for new methodologies that employ a multi-model combination approach by including several scenarios for improving decision making. A multi-model combination is based on the results of several models and creates a more robust forecasting system with a better representation of uncertainties (Kauffeldt et al., 2016). Although the multi-model combination approach has been used widely in the field of discharge forecasting (Shamseldin et al., 1997; See and Openshaw, 2000; Oudin et al., 2006; Weigel et al., 2008), the approach is not commonly used in the field of

real-time flood hazard forecasting. The high-computational time required by the HD models restricts the use of such an approach in real-time forecasting. However, the use of a simple model structure and/or high-performance computing makes it possible to simulate HD models in real-time; thus, making it feasible to use multi-model combination approaches. Zarzar et al. (2018) have used a multi-model combination framework consisting of hydro-metrological and HD models to visualise flood inundation uncertainties in which they have used an average of HD model raster outputs to obtain the percentage of ensemble agreement.

We develop a methodology for obtaining a multi-model combination as an effective alternative to traditional *best-model* approach for producing detailed hazard maps, which are termed as *building hazard maps*. This term can be defined as a map that highlights buildings that are affected by or are vulnerable to flooding with differentiated exceedance probabilities of flood inundation extents projected on building use. In this manuscript, we have designed three scenarios with differentiated exceedance probabilities, each referring to the subjective classification of buildings with varying flood impact. To the best of our knowledge, this combination approach has yet not been used to assess the impact of flooding. The maps help prevent serious damage to buildings and aid in evacuation planning in the case of flooding. The methodology is applied for the flood event of January 2011 in the city of Kulmbach, Germany.

## 2 Methodology

The framework to generate *building hazard maps* (as shown in Figure 1) consists of three components: (1) Hydrological modelling – discharge ensemble forecasts were produced using forecasted precipitation; (2) HD modelling – the water depths were simulated using a pre-calibrated 2D HD model; (3) Post-processing of the model results – a multi-model combination was used to produce flood hazard maps based on a classification of buildings. The framework was tested for the flood event of January 2011 in the city of Kulmbach, Germany. The first two components of the framework were developed in previous studies (Beg et al., 2018; Bhola et al., 2018a, Bhola et al., 2018b). The particular focus of this study is on the development of the framework of a multi-model combination in the post-processing component. For the sake of clarity, each component is described in detail in chronological order.

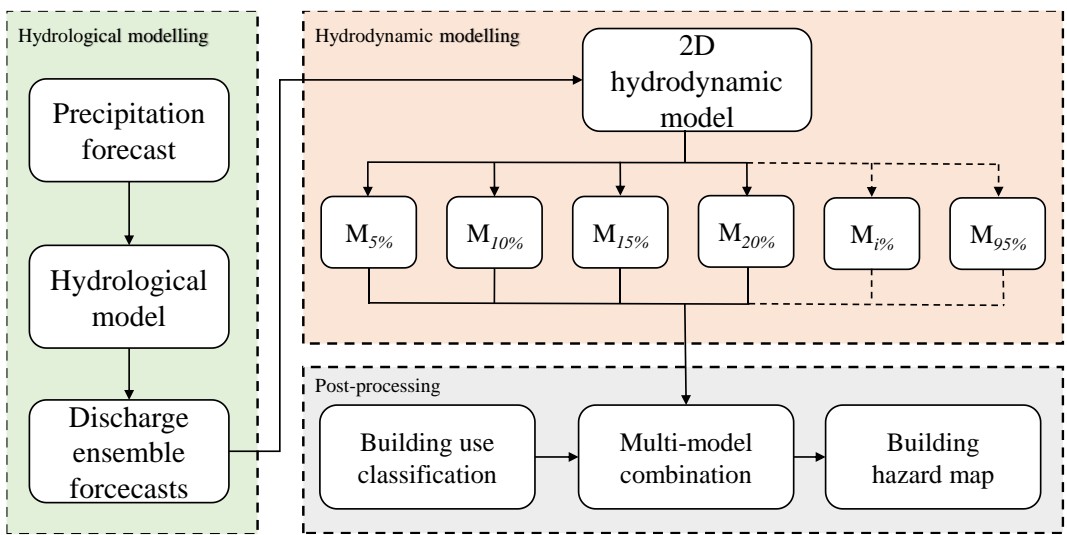

**Figure 1. Schematic view of the methodology used to generate building hazard maps. The major components consist of the operational hydrological ensemble forecasts (Beg et al., 2018), the hydrodynamic model and post-processing that includes the multi-model combination. $M_{x\%}$ denotes the HD model results generated using x% percentile discharge.**

### 2.1 Hydrological modelling

5    2.1.1 Hydrological model - LARSIM

The conceptual hydrological model LARSIM (Large Area Runoff Simulation Model) was used to study the hydrology of the model area and to generate discharge forecasts. In the model, the hydrological processes are simulated in a series of subarea elements connected by flood routing elements in a pre-determined sequence. LARSIM simulates the hydrologic processes for one element for a defined period and passes the resulting output hydrographs information to the next element (Figure 2). The

10    model structure can be both grid-based or based on hydrologic sub-catchments. The model uses a soil module with storage capacities in considering infiltration, evapotranspiration and runoff generation. The discharge generation consists of three components: runoff generation, runoff concentration and river component. In addition to simulating hydrological processes, LARSIM is most suitable in operational flood forecasting (Demuth and Rademacher, 2016). It deals with the gaps in hydrometeorological input data and allows for the correction/manipulation of numeric weather forecasts (e.g. external forcing

15    parameters). Furthermore, the model automatizes processes for the assimilation of hydrological data, which is crucial in flood forecasting (Luce et al., 2006; Haag and Bremicker, 2013).

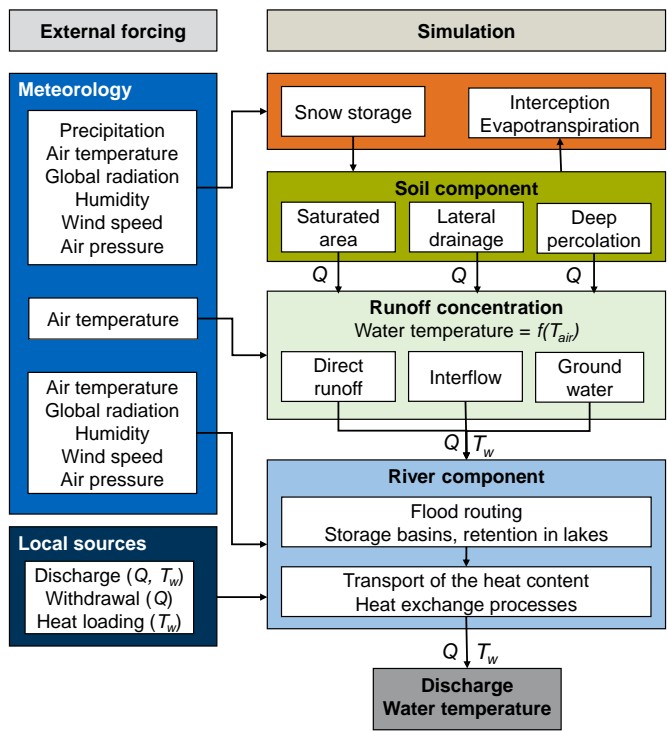

**Figure 2. LARSIM water balance model. Source based on Ludwig and Bremicker (2006).**

For this study, a pre-setup model for the study area was provided by the Bavarian Environment Agency and this model is operationally used in the Flood Forecasting Centre for the river Main (Laurent et al., 2010). The model uses a grid-based structure with a resolution of 1 km$^2$ and a temporal resolution of 1 hour. This LARSIM model considers a soil module with storage capacities in considering the water balance, which consists of three parts: upper, middle and lower soil storages that contribute to the discharge components, modelled as a linear storage system. The model includes 34 parameters that allow the modelling of different processes, such as direct discharge, interflow and groundwater flow. A complete description of calibration parameters is not the scope of this study and has been elaborated on by Ludwig and Bremicker (2006) or Haag et al. (2016). Nevertheless, Table 1 in the Supplement presents is provided for a comprehensive description of important parameters along with eight most sensitive parameters identified in Beg et al. (2018), which were considered in generating the discharge ensemble forecasts.

### 2.1.2 Discharge ensemble forecasts

The winter flood event of January 2011 was hindcasted to test the framework. The event was one of the largest in terms of its magnitude and corresponds to a discharge of 100-year return period at gauge Kauerndorf (river Schorgast) and 10-year return period at gauge Ködnitz (river White Main). Intense rainfall and snow melting in the Fichtel mountains caused floods in several rivers of Upper Franconia. Within five days, two peak discharges were recorded. The first peak occurred on 9th January 2011,

and the second peak measured five days later (on 14[th] January 2011) caused even higher discharges and water levels. The maximum discharge of 92.5 m³/s was recorded at gauge Kauerndorf and 75.3 m³/s at gauge Ködnitz (Figure 3).

To automatize the generation of forecasts, a tool *FloodEvac* was developed in MATLAB® R2018a (Disse et al., 2018). The tool considers model input and model parameter uncertainty in simulating flood scenario combinations. The tool generates
rainfall spatial distributions using sequential conditional geospatial simulations and model parameter uncertainty using Monte-Carlo sampling. The uncertainties in the discharge hydrographs were quantified in Beg et al. (2018) using this FloodEvac tool. In their study, the forecast was performed using 50 ensemble members. Parameter uncertainty module was used to generate 50 different parameter sets (for eight sensitive parameters). In addition, geostatistical simulation for rainfall was implemented using two different R-packages, namely *gstat* and *RandomFields*. The rainfall data was available at an hourly interval at 50
gauges in the catchment. Each forecast was simulated for 61 hours: 49 hours of observed hourly rainfall and 12 hours of forecast rainfall data. To hindcast the event of January 2011, 10 different raster dataset of rainfall uncertainty was generated for the catchment. The 50 parameter sets were combined with the 10 rainfall uncertainty cases, linking one rainfall scenario with every 5-parameter sets in sequential order, thus, making 50 sets of hydrological models for the Upper Main catchment. These 50 models were then simulated, and the results of discharge ensembles were stored.

Figure 3 shows the percentiles of 10%, 25%, 50%, 75% and 90% for the January 2011 flood event at two gauging stations upstream of the city, Ködnitz and Kauerndorf. Uncertainty bands are much wider at gauge Ködnitz (Figure 3a) than at gauge Kauerndorf, which suggests that the model parameters are more sensitive in the catchment of White Main than Schorgast. In addition, the peak of the measured discharge at gauge Ködnitz was well over-predicted, which suggests that the uncertainty of the discharges is higher in the catchment of White Main than Schorgast. While the peak of the measured discharge at
Kauerndorf is very well predicted, the one at the gauge Ködnitz is over-predicted. Nevertheless, it can be seen from Figure 3 that the ensemble of these 50 members could effectively bracket the observed discharge data.

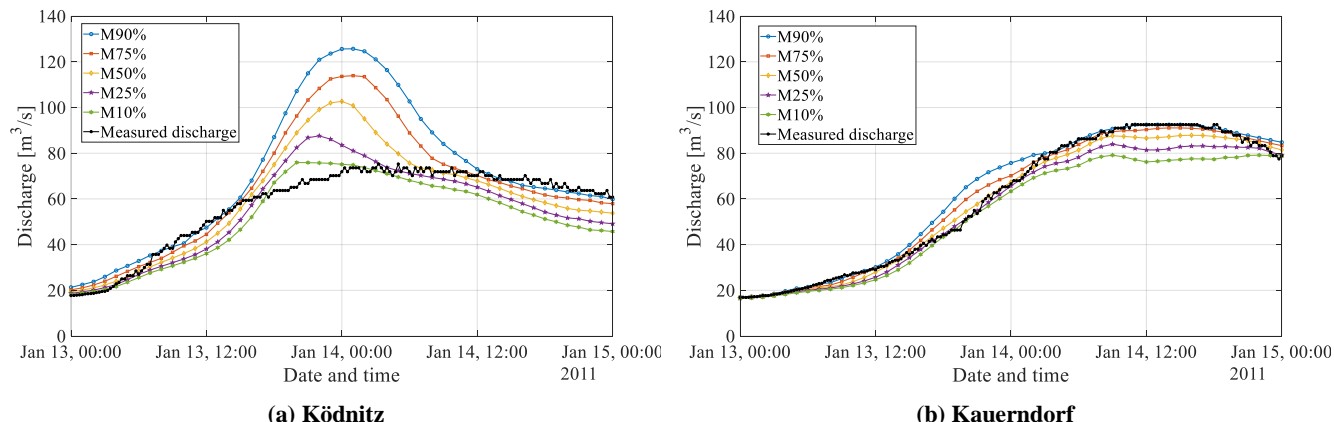

(a) Ködnitz                                                          (b) Kauerndorf

**Figure 3. Hindcasted flood event of January 2011: measured discharge hydrograph along with 10%, 25%, 50%, 75% and 90% percentile discharges for gauges a) Ködnitz and b) Kauerndorf (Discharge data based on Beg et al., 2018; Measured discharge from Bavarian Hydrological Service, www.gkd.bayern.de, last access: 5 March 2018).**

## 2.2 Hydrodynamic modelling

HEC-RAS was used as the 2D HD model to quantify uncertainties in flood inundation. It is a non-commercial hydrodynamic model developed by the U.S. Army Corps of Engineers and has been used widely for various flood inundation applications (Moya Quiroga et al., 2016; Patel et al., 2017). The implicit method allows for larger computational time steps compared to
an explicit method. HEC-RAS solves either 2D Saint Venant or 2D diffusion-wave equations. The latter allows faster calculation and has greater stability due to its complex numerical schemes (Martins et al., 2017). Due to these advantages and suitability for use in real-time inundation forecast (Henonin et al., 2013), we have used the diffusive-wave model that was previously set-up, calibrated and validated in Bhola et al. (2018a) and Bhola et al. (2018b). For the diffusive-wave approximation, it is assumed that the inertial terms are less than the gravity, friction, and pressure terms. Flow movement is
driven by a barotropic pressure gradient balanced by bottom friction (Brunner, 2016). The equations of mass and momentum conservation are as follows:

$$\frac{\partial H}{\partial t} + \frac{\partial (hu)}{\partial x} + \frac{\partial (hv)}{\partial y} + q = 0 \qquad (1)$$

$$g\frac{\partial H}{\partial x} + c_f u = 0 \qquad (2)$$

$$g\frac{\partial H}{\partial y} + c_f v = 0 \qquad (3)$$

$$c_f = \frac{g|V|}{M^2 R^{4/3}} \qquad (4)$$

Where H is the surface elevation (m); h is the water depth (m); u and v are the velocity components in the x- and y- direction respectively (ms$^{-1}$); q is a source/sink term; g is the gravitational acceleration (ms$^{-2}$); cf is the bottom friction coefficient (s$^{-1}$); R is the hydraulic radius (m); |V| is the magnitude of the velocity vector (ms$^{-1}$); and M is the inverse of the Manning's n (m$^{(1/3)}$s$^{-1}$).

Table 2 in Supplement summarises the model properties, such as the model size and mesh size, and model roughness in the domain. The model parameter consists of the roughness coefficient Manning's M for five land use classes. The buildings are explicitly included using their shape in the mesh and are excluded from the flow calculation by assigning a high roughness value. To assign hazard to a building, the maximum water depth of all the neighbouring cells was used. Sensitivity analysis of the model was performed using one thousand uniformly distributed model parameter sets for the flood event of 2011.

Although uncertainties arise in the HD modelling, we have considered discharges in hydrological modelling as the sole source of uncertainties in this paper as we have assumed them to be more significant. Various HD simulations were conducted based on percentiles of the discharges (Figure 3) as upstream boundary conditions at river gauges Ködnitz and Kauerndorf.

## 2.3 Post-processing

### 2.3.1 Building use classification

In this study, we have considered only buildings as urban features to access the flood impact and in preparation of flood hazard maps. The shape and use of the buildings were provided by the Bavarian Ministry of the Interior, for Building and Transport (Figure 4).

There are various classifications of land use features available in the literature. Dutta et al. (2003) have used direct and indirect damages as the basis of their classification and classified their study area in residential and non-residential categories. Jonkman et al. (2008) have classified urban features in residential, businesses, commercial and public property and agricultural to estimate flood loss. Furthermore, the vulnerability was the basis of classification in residential (Thieken et al., 2008) and industrial and commercial sectors (Kreibich et al., 2010) in order to estimate flood losses. We have used the damage potential of a building as a basis for classification in order to focus on the flood impact assessment. Building damage potential is required for a variety of flood mitigation planning activities including flood damage assessment, multi-hazard analyses and emergency measures (Shultz, 2017). The buildings were classified into four classes based on their function following the recommendation of the German standard for risk management in urban areas in the case of flash floods (Krieger et al., 2017). According to this standard, building use is one of the important criteria for assessing the damage potential of a building. In this study, four damage potential classes for each building use were taken into consideration as presented in Table 1. In the authors' opinion by keeping our classification simple will likely fit a vast majority of cities regardless of their size. In any case, we acknowledge that the number of classes or criteria can be changed/adapted depending on the aim of the forecast.

The damage potential varies from *low* to *very high* based on the building use, for example, residential buildings with a basement, industries and schools need special protection and thus were rated with a correspondingly high damage potential (class III). In addition, nursery, hospitals as well as low-lying facilities, such as traffic underpasses, driveways to underground garages and other entrances require greater protection and are thus categorised as having the highest damage potential (class IV). Residential buildings and retail businesses were classified as having moderate damage potential (class II), and gardens and parks relatively *low* damage potential (class I). Figure 4 shows the city centre, where buildings were classified according to Table 1. It can be seen that most of the buildings belong to class III as the area is industrial. There are a total of 2695 buildings in Figure 4 of which 1, 958, 1716 and 20 buildings were classified in classes I, II, III and IV respectively. The nature of the data in this case study leads to a differentiated representation of the classes. It should be noted that the classification aims at creating classes based on damage potential, and not on generating clusters with similar sizes.

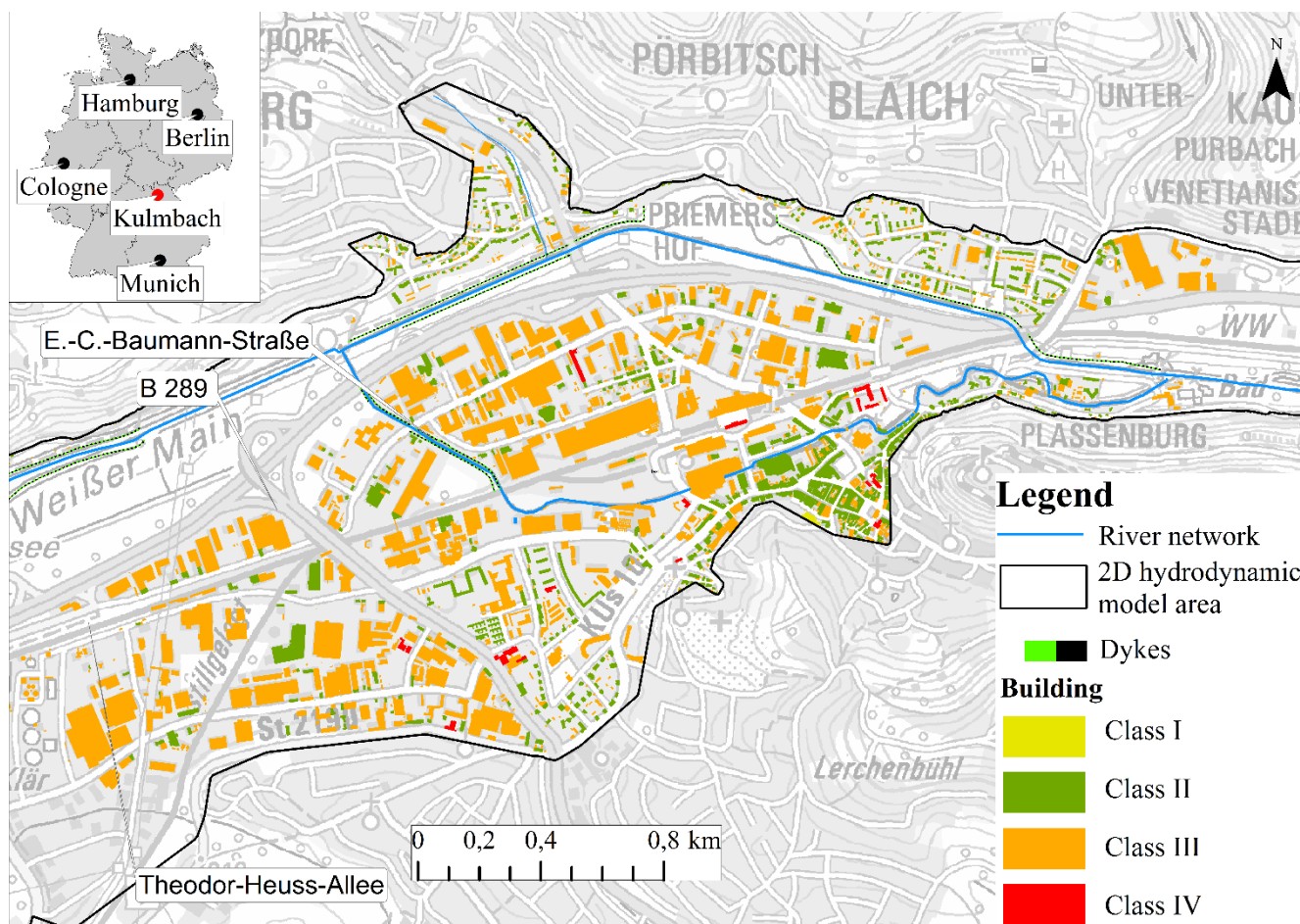

**Figure 4. The city of Kulmbach and building use classification. (Data source: Bavarian Ministry of the Interior, for Building and Transport and Water Management Authority Hof; Geobasisdaten © Bayerische Vermessungsverwaltung, www.geodaten.bayern.de, last access: 5 March 2018).**

**Table 1. Building use classification based on the guidelines of Krieger et al. (2017).**

| Class | Building use | Damage potential |
|:---:|:---:|:---:|
| I | Garden buildings <br> Parks / green areas | low |
| II | Residential building without a basement <br> Retail / small business | moderate |
| III | Residential building with basement (inhabited) <br> Industry / Trade <br> School / College | high |
| IV | Nursery / hospital / nursing home / emergency services <br> Energy / telecommunications <br> Underground car park <br> Metro access / Subways | very high |

### 2.3.2 Hazards classification

In this study, hazard classification was based on the recommendations given in the German standard for risk management in urban flood prevention (Krieger et al., 2017). The classification was based on the estimated water depths of the 2D HD model. Table 2 shows the four categories of flooding hazards, which consider water flow in urban areas and vary from *low* to *very high*. It should be noted that in individual cases, the risk may also arise at lower water depths (<0.10 m) for buildings, such as underground parking and metro stations, which are classified as the building class IV in the previous section.

**Table 2. Hazard classification used in this study based on water depths. Classification source Krieger et al. (2017)**

| Hazard class | Flooding hazard | Water depth [m] |
|:---:|:---:|:---:|
| 1 | low | < 0.10 m |
| 2 | moderate | 0.10 – 0.30 m |
| 3 | high | 0.30 – 0.50 m |
| 4 | very high | > 0.50 m |

### 2.3.3 Multi-model combination

The multi-model combination of the 2D HD model results was based on considerations of evacuation planning and gives priority to buildings with higher damage potential. In order to prioritise, it is important to differentiate the impact of water depths on building classes. A certain water depth might have a different impact on a building with higher damage potential. For example, there is more threat for a low water depth in underground metro access that the same water depth to a residential

building. Therefore, buildings classified to higher damage potential class relates to model results of a higher percentile. Each building class corresponds to a certain discharge percentile and the resulting damage potential assessment can be visualised and presented as a building hazard map.

Figure 5 shows an example of a multi-model combination in which the four building classes were assigned four different percentiles. The simulation results (water depth in this case) obtained from the HD model with 25%, 50%, 75% and 90% percentile discharges were assigned to the building classes I, II, III and IV respectively. The novelty of the multi-model combination approach is that the flood inundation uncertainty is coupled with the building use. As such evacuation planning or investment planning can take the information of uncertainties in the water depths into consideration.

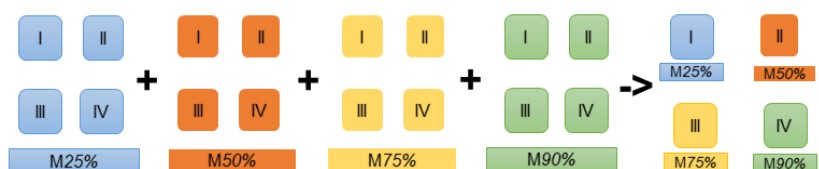

**Figure 5. An example of a multi-model combination in which the four building classes I, II, III and IV are assigned to the 2D HD model results of 25%, 50%, 75% and 90% respectively.**

## 3 Results

In this section, we present the results of five percentiles and the performance of the multi-model combination. To assess the methodology, the flood event of January 2011 was used to quantify uncertainties in discharge hydrographs. The forecasts corresponding to 10%, 25%, 50%, 75% and 90% percentiles were further used as input boundary conditions to the 2D HD model and water depths were stored. Furthermore, the flood inundation maps and building hazards were then classified.

### 3.1 Flood inundation maps and building hazards

The number of affected buildings in each hazard class for all five HD models are presented in Figure 6. As the discharge percentile increases, the number of affected buildings in each hazard class increases. The maximum flood inundation of the five models is presented in Figure 7. The figures present both the inundation extent and building hazards based on the classification discussed in section 2.3.2.

Post-event binary information of the flood extent was collected from newspaper articles and press releases published by the Bavarian Water Authority. The information shows that the dykes were at their full capacity and most of the floodplains and traffic routes were flooded, but no serious damage was reported (Hof, 2011). The streets Theodor-Heuss-Allee and E.-C.-Baumann-Straße were flooded and some flooding was observed on motorway B289 (see Figure 4 for locations).

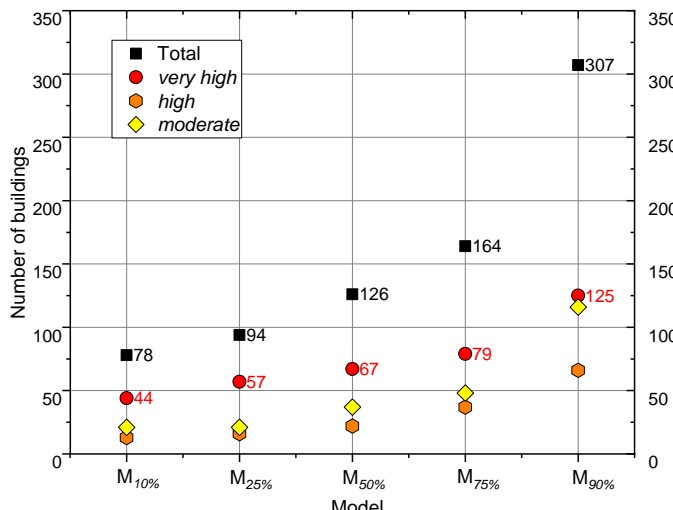

**Figure 6. The number of affected buildings in each hazard class for 2D HD model results using five discharge percentiles.**

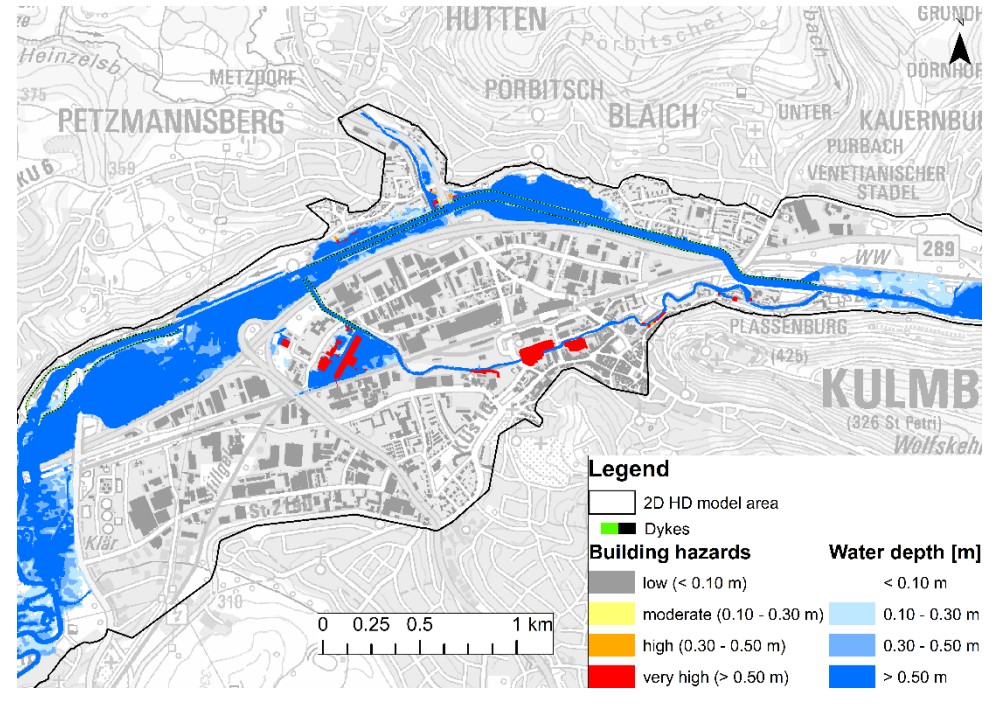

**(a) M$_{10\%}$**

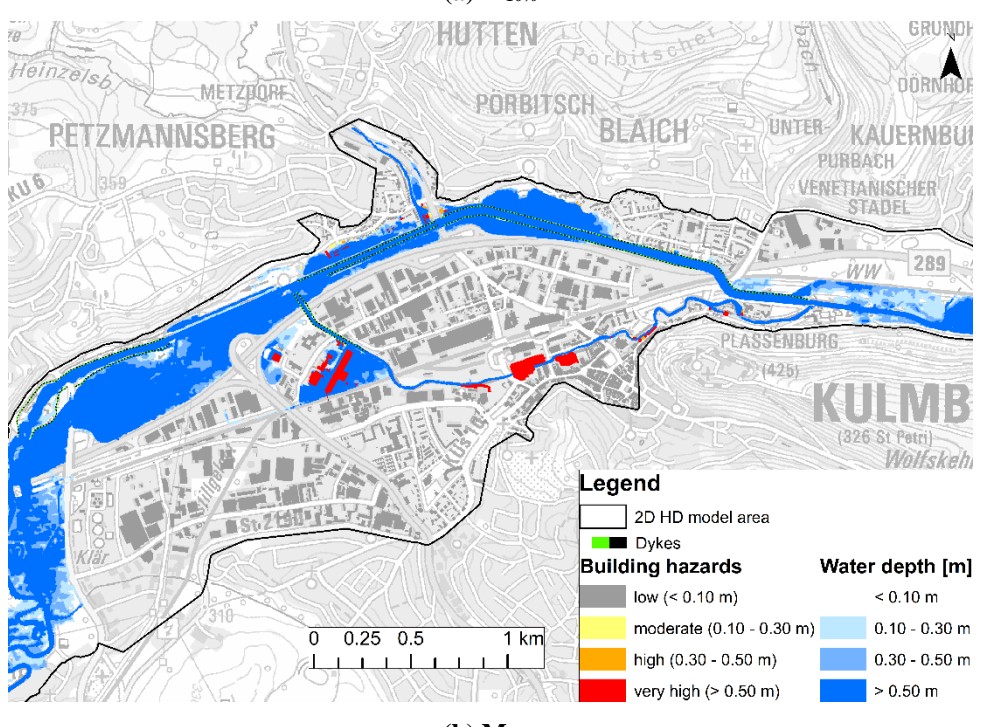

**(b) M$_{25\%}$**

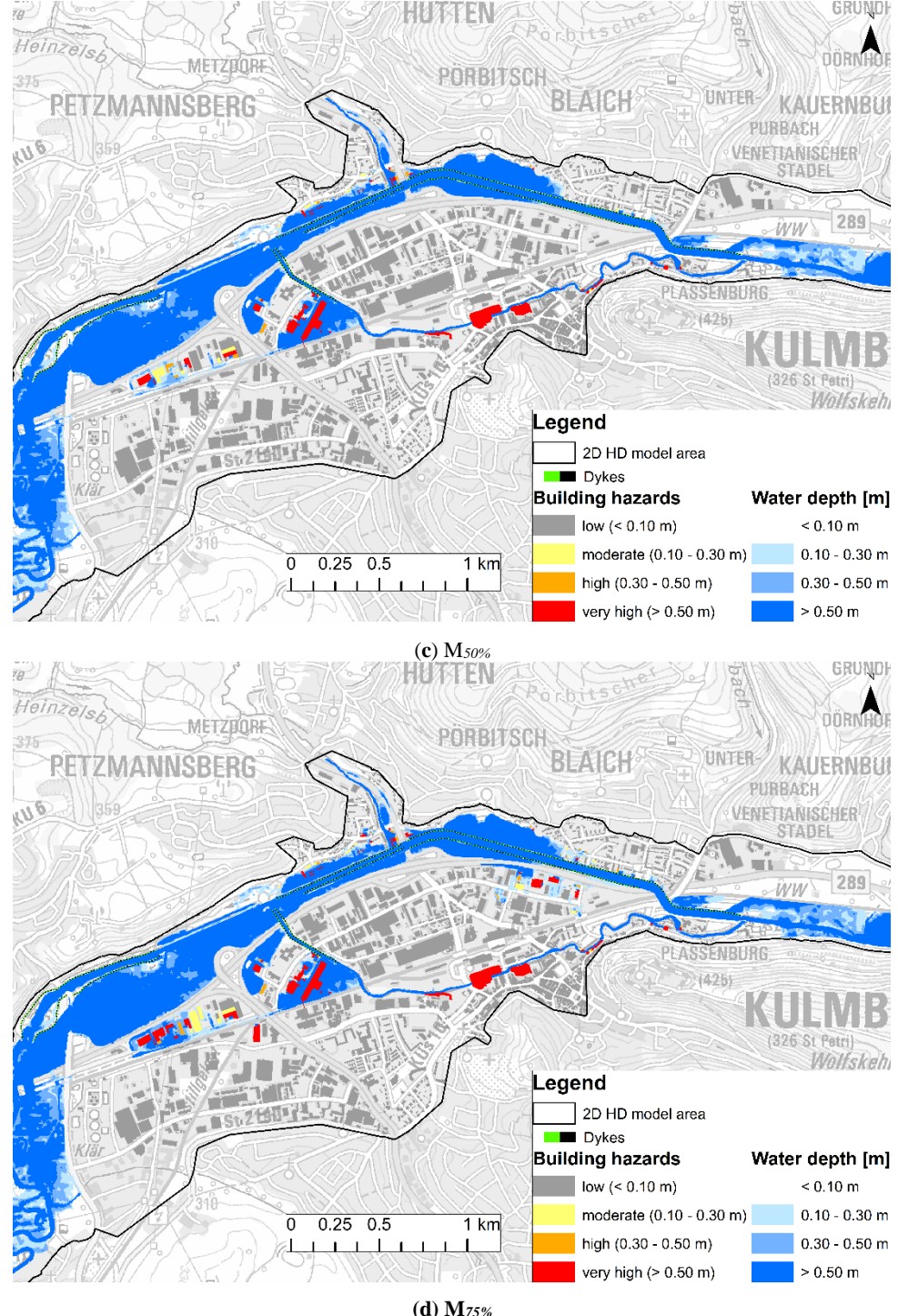

**(c)** M*50%*

**(d)** M*75%*

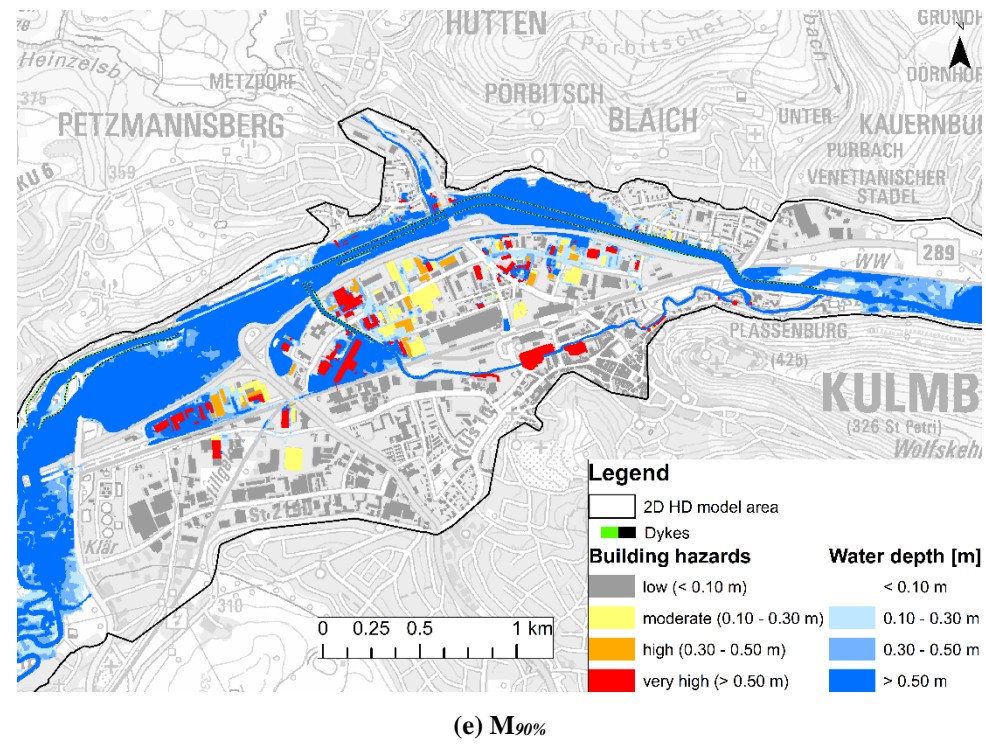

**(e) M$_{90\%}$**

**Figure 7. Flood inundation and building hazard maps for five percentiles discharge hydrographs. (Data source: Geobasisdaten © Bayerische Vermessungsverwaltung, www.geodaten.bayern.de, last access: 5 March 2018)**

### 3.2 Multi-model combination

Three combination scenarios based on *high*, *average* and *low* exceedance probability were designed to illustrate the methodology developed in this study and are presented in

Table **3**.

**Table 3. Scenarios of multi-model combinations based on exceedance probability.**

| Scenario | Exceedance probability | Building class | | | |
|:---:|:---:|:---:|:---:|:---:|:---:|
| | | I | II | III | IV |
| **I** | High | M$_{10\%}$ | M$_{10\%}$ | M$_{25\%}$ | M$_{50\%}$ |
| **II** | Average | M$_{10\%}$ | M$_{25\%}$ | M$_{50\%}$ | M$_{75\%}$ |
| **III** | Low | M$_{25\%}$ | M$_{50\%}$ | M$_{75\%}$ | M$_{90\%}$ |

The main objective of the combination is to differentiate the impact of water depths on building classes. Therefore, to design the combinations, a high percentile was assigned to the buildings with a high damage potential class. Each scenario presents a given risk perception that can be defined as the subjective judgement of a decision-maker about the severity of the risk, which can influence the choice of mitigation measures (Botzen and van den Bergh, 2009). Different risk perceptions will lead to

different exceedance probability scenarios, which can be easily adjusted depending on the perception of different stakeholders. The hazard maps for the three scenarios are shown in Figure 8.

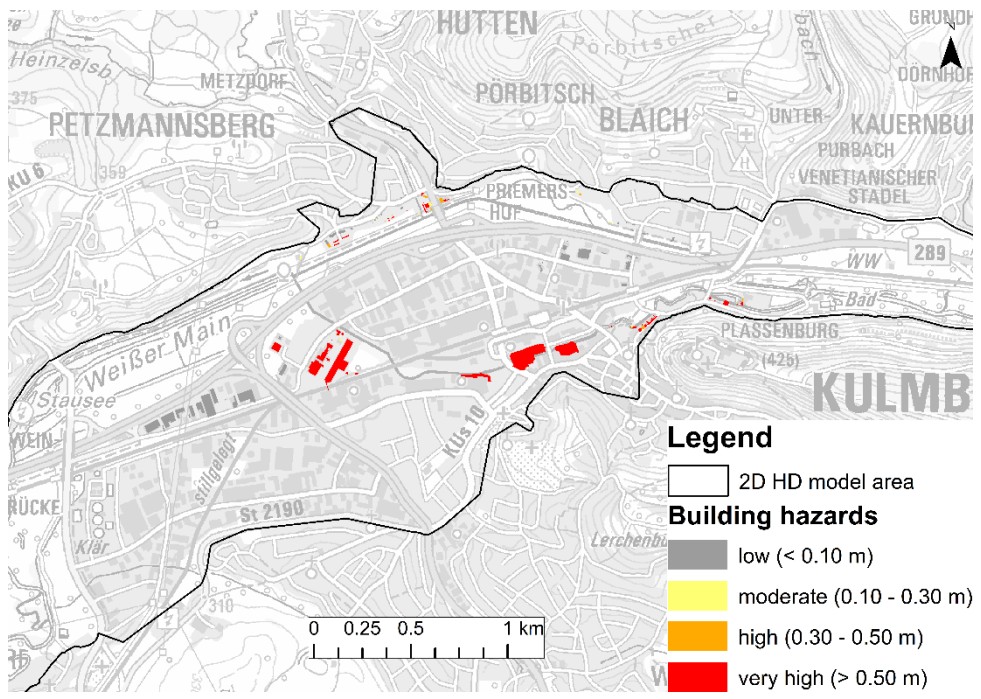

**(a) Scenario I:** *high* **exceedance probability**

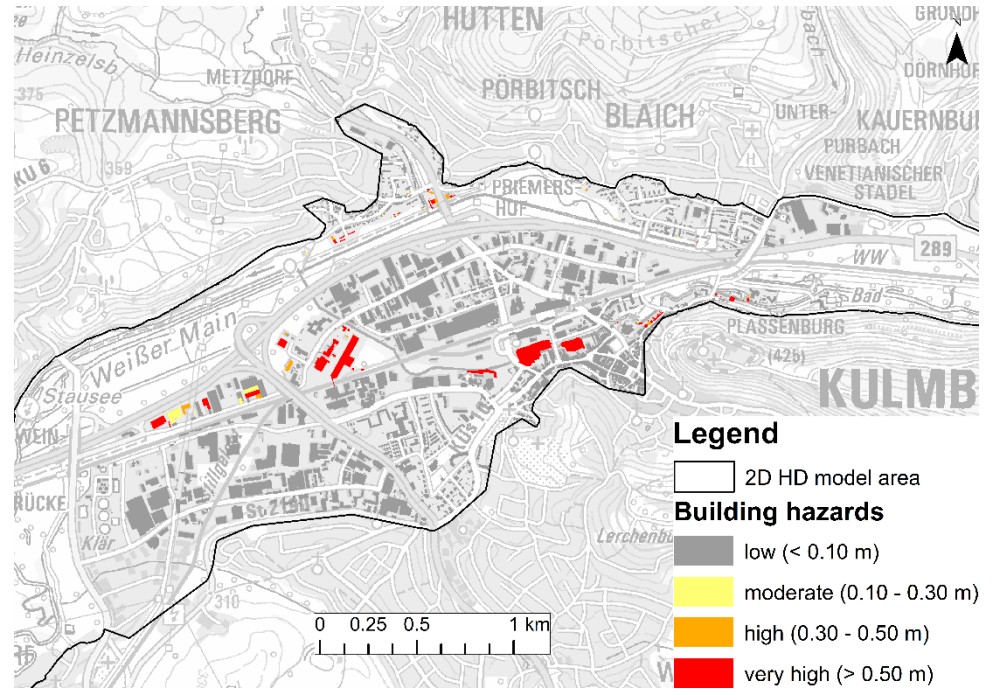

**(b) Scenario II:** *average* **exceedance probability**

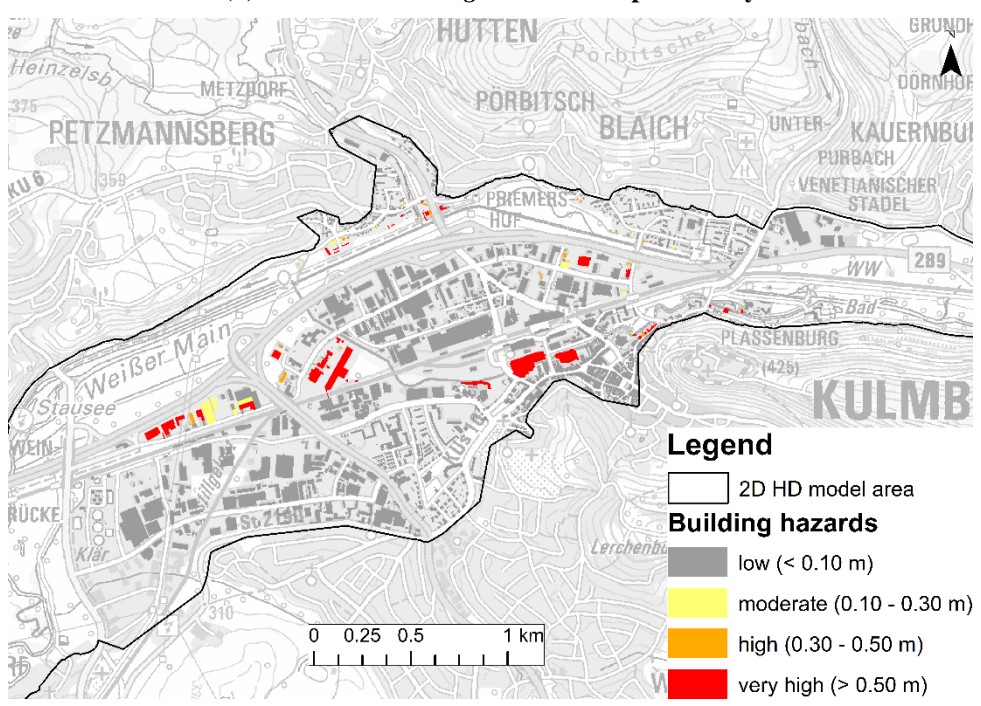

**(c) Scenario III:** *low* **exceedance probability**

**Figure 8. Building hazard maps for the three scenarios, the numbers of affected buildings are 84, 107 and 142 respectively. Hazard classification is based on Krieger et al. (2017). (Data source: Geobasisdaten © Bayerische Vermessungsverwaltung, www.geodaten.bayern.de, last access: 5 March 2018)**

## 4 Discussion

Prior work in hydrology has demonstrated the effectiveness of multi-model combinations in improving flood forecasts as compared to the *best-model* approach (Weigel et al., 2008). However, these methodologies were previously limited to discharge ensemble forecasts and were not researched for hazard maps. In this study, we extend the use of multi-model combinations to produce flood hazard maps for buildings depending on their use.

First, the five simulation results are presented in Figure 7 as inundation and building hazard maps. It should be noted that few buildings show *very high* hazards due to their proximity to the Mühl canal (Figure 7a). Even though there was no over-topping of water from the canal, because of buildings geolocation being near to the canal, these were assigned automatically with the highest hazard, starting with a discharge of $M_{10\%}$. Ideally, this should be prevented by removing the river channel elements from the dataset before to assign the water depths to the buildings as in Bermúdez and Zischg (2018). However, and without retract to our conclusions, it was decided not to include it in this work in order to keep the automation process simple. Up to a discharge of $M_{50\%}$, no inundation in the city centre was observed as the dykes were not breached. It can be observed in Figure 6, that the increment in the number of affected buildings is gradual, especially the buildings belonging to *very high* hazard class. As the peak discharge increases in $M_{75\%}$, the dykes at the B289 road were breached and water entered in the city centre and more buildings were affected. Most damages were observed in $M_{90\%}$ with 307 affected buildings, out of which 125 buildings show *very high* hazard, an increment of 46 from $M_{75\%}$. The affected buildings were located in the city centre (Figure 7e), mainly in industrial and commercial areas. Similarly, the streets Theodor-Heuss-Allee and E.-C.-Baumann-Straße were inundated starting from a discharge of $M_{50\%}$.

In operational use, the mean of the discharge ensemble or $M_{50\%}$ would normally have been used as the *best-model*, which according to Figure 7c, is in agreement with the post-event information. However, this match might not always be representative, especially in the case of an event of different or higher magnitude, as discussed in Di Baldassarre et al. (2010). They argued that visualising flood hazards as a probability is a more accurate representation as compared to a single *best-model*, which might misrepresent the uncertainty in the modelling process.

With the objective of visualising uncertainties, three scenarios based on exceedance probability were used to combine HD model results and are presented in Figure 8. In scenarios I and II, 84 and 107 buildings were affected, which shows that the impact of *high* and *average* exceedance probability scenarios was less as compared to $M_{50\%}$ in which a total of 126 buildings were affected, out of which 67 buildings were classified in *very high* hazard class.

Furthermore, as the majority of the buildings were classified in class II and III, the resulting map of a *low* exceedance probability scenario corresponds closely with $M_{50\%}$ and $M_{75\%}$, with 142 affected buildings. In scenario II, 63 buildings were classified in the *very high* hazard class, which increased to 71 in scenario III. Similarly, 22 buildings belonged to both *moderate*

and *high* hazard classes, and shifting to scenario III, the number increased to 33 and 38 in the *moderate* and *high* classes respectively.

In Figure 9, a comparison is presented between the *best-model* ($M_{50\%}$) and the multi-model combinations and the areas with prominent changes are highlighted in red circles. The figure presents building hazards resulting from the combination of exceedance probability scenarios and locates 16 buildings that are affected as compared to $M_{50\%}$. The buildings that belong to class III (Figure 9b) were assigned the results of $M_{75\%}$, and show a *very high* hazard. Figure 9a shows that an adjacent building belonging to class II (ID 1393) was not flooded. This demonstrates that the methodology was implemented accurately and prioritised measures such as flood impact assessment, spatial planning, early warning and emergency planning, according to the damage potential of a building. The prioritisation is important in order to focus on a combination of various evacuation strategies to prevent damage and save lives (Kolen et al., 2010). Hence, decision-makers must be made aware of the impact associated with a *low* exceedance probability to improve their planning strategies (Pappenberger and Beven, 2006; Uusitalo et al., 2015).

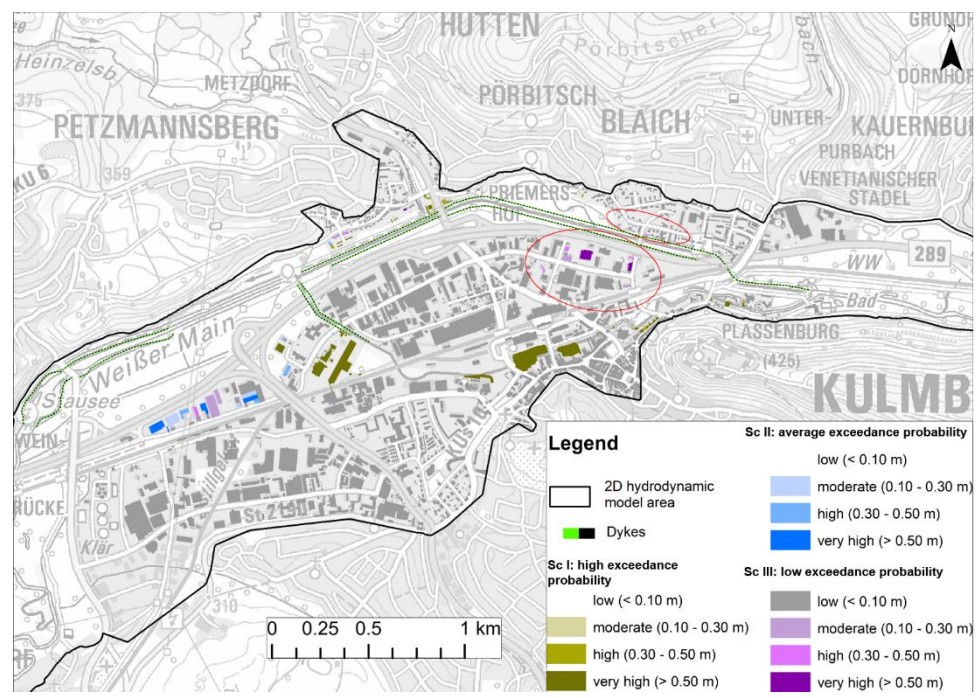

**(a) Multi-model combination**

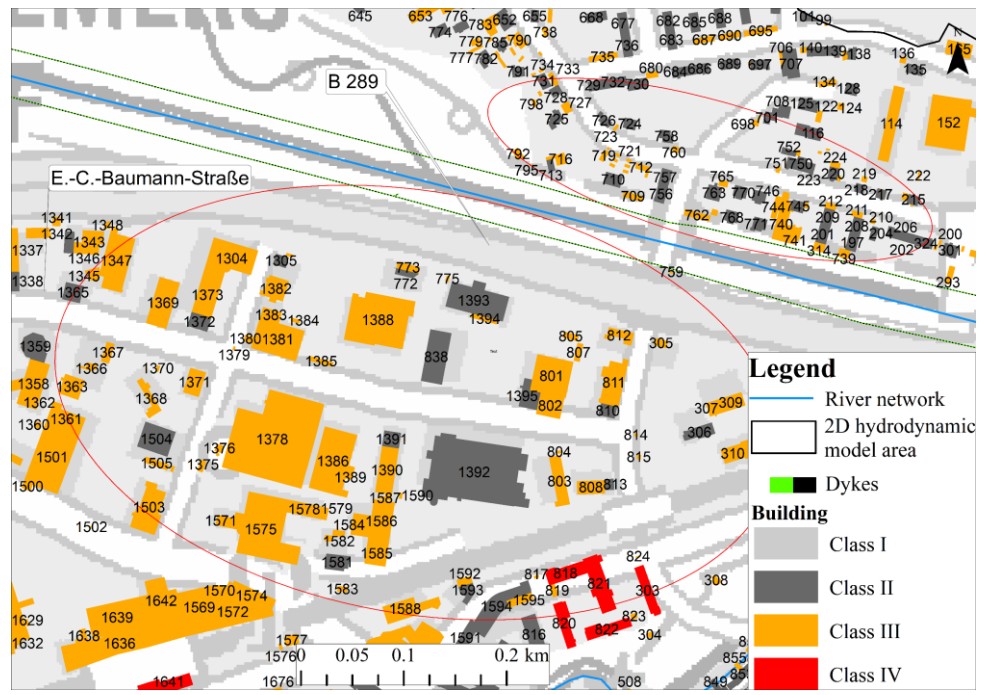

**(b) Building use classification**

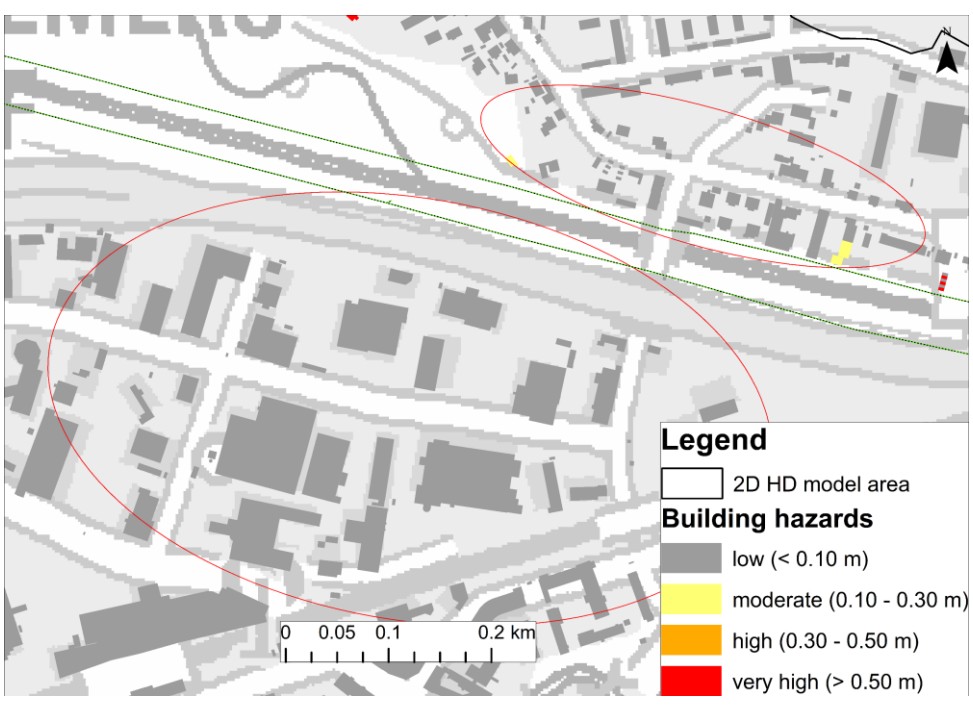

**(c) M**$_{50\%}$

**Figure 9. Comparison of building hazard maps between *best-model* (M$_{50\%}$) vs. multi-model combinations. The areas with prominent change are highlighted in red circles. (Data source: Geobasisdaten © Bayerische Vermessungsverwaltung, www.geodaten.bayern.de, last access: 5 March 2018)**

A potential drawback of the combination is that the hazard classification may shift from *low* to *very high* in two adjacent buildings belonging to different classes. This might confuse evacuation planners by presenting inconsistent information. To tackle this issue, more information and specific guidelines should be provided to them on how to use the maps. In addition, continuous flood inundation maps are hard to obtain, especially at the boundaries of two combinations. There might be a step rise in the water depths while shifting from the results of one model to another. To address this issue, future research should be conducted to provide consistency in interpolation and in combining models (see Zazar et al., 2018). In addition, in order to avoid the confusion, these maps could be forecasted for a regular interval of 3-4 hours.

Overall, the methodology is independent of the choice of models, i.e. hydrological and HD, and is transferable to other study areas. In order to use the methodology in real-time, the run-time of the flood forecasting modelling should be below the flow travel time. In this study, a 50-member ensemble forecast was used from Beg et al. (2018) where the entire process took 25 minutes in a 3-core desktop in parallel mode to generate a forecast of 12 hours. Various percentile discharges were then run simultaneously in the HD model, which required 30 minutes to simulate a 12-hour event on an 8-core, 2.4 GHz (Intel E5-2665), including the initial start (Bhola et al. 2018a). Post-processing of the model results would consume an additional 15 min. Therefore, real-time hazard maps are delivered to decision-makers in 70 minutes. A faster run-time can be ensured by either using a simple model structure (Leandro et al., 2014) and/or high-performance computing (Kuchar et al., 2015). In the absence of such infrastructures or a very large catchment size, HD models can be replaced with alternatives, such as terrain-based models (Zheng et al., 2018) and satellite images (Voigt et al., 2007). In addition, a database of pre-recorded inundation scenarios as shown in Bhola et al. (2018a) can expand the application of this methodology.

Molinari et al. (2014) have stated that a comprehensive uncertainty assessment improves emergency responses by assessing the potential consequences of flood events. Therefore, our methodology would allow the target users to benefit from hazard maps enabling them to better prioritise and coordinate evacuation planning based on the stakeholder perception to risk. The maps could further serve as a tool for flood risk assessment. The methodology can be used for flood mitigation and flood forecast planning in the form of emergency management training, where forecasted hazard scenarios can be presented to the training groups. By visualising inundation scenarios, potential damage at the building's level which has been prioritized based on the desired classification can be estimated with this methodology and made available together with each forecasted scenario.

## 5 Conclusions

In summary, we have presented a new methodology for flood impact assessment using a multi-model combination in the form of *building hazard maps with differentiated exceedance probability*. These maps offer an alternative way to communicate the underlying uncertainties in forecasting models and are ready-to-use for decision-makers in the field of flood risk management. The entire forecasting framework consists of three stages: (i) generation of discharge ensemble forecasts, (ii) 2D HD

simulations using the generated forecasts and (iii) hazard maps using multi-model combinations. The framework was applied to the city of Kulmbach and three multi-model combinations were designed based on exceedance probability. The model results of $M_{50\%}$ show a good match with binary information collected after the flood event. The *low* exceedance probability scenario corresponds closely with $M_{50\%}$ and $M_{75\%}$. We expect this multi-model combination to improve the current

visualisation techniques in operational flood risk management and evacuation planning. In this study, we have considered only buildings as a feature; additional urban features, such as bridges (Gebbeken et al., 2016) and roads (Goerigk et al., 2018), should be included in future to extend the methodology. Furthermore, other sources of uncertainty, such as HD model parameters, model structures and measured data should also be incorporated for a comprehensive assessment. In addition, the economic, social and hazardous effects of carrying out an evacuation in the case of false alarm must be considered. Hence, a

validation of the combination is crucial to building trust in its prediction in real-time. Further research investigating multi-model combinations and validation in other study areas may be beneficial. In order to design a multi-model combination, a group consisting of researchers, operational bodies and experts in the field of flood risk management should be consulted. A more extensive study on the validation of the multi-model combination may be required, possibly by using measuring gauges, post-event survey (as conducted in Thieken et al., 2005), satellite images (as in Triglav-Čekada and Radovan, 2013), and/or

crowd-sourced data (Bhola et al., 2018b).

In future, damage potential classification can further be improved by including additional criteria, such as population density or water quality, and with it extend the applicability of this method. For example, the assessment of the damage potential of commercial enterprises, substances or machinery containing elements that which could be a source of water pollution could be included (Krieger et al., 2017). In addition, other classification methods for buildings and hazard types should be evaluated,

especially to further dissect the impact of class III in commercial and industrial. Finally, the output of the framework can be extended to hazard maps uploaded in a web-based GIS system to improve visualization, along with providing layers of additional information, such as inundation pathways and weak spots in the river and floodplains to provide sufficient details to intervene (aid in planning). This additional information would enhance the usefulness to different target users, such as planners, decision-makers and flood forecasting agencies.

**Author contribution**

Punit K. Bhola conceptualised and completed the formal uncertainty analysis. Punit K. Bhola wrote the original draft and subsequently reviewed and edited by all co-authors. All authors contributed to writing the paper.

**Competing interests**

The authors declare that they have no conflict of interest.

## Acknowledgements

This research was funded by the German Federal Ministry of Education and Research (BMBF) with the grant number FKZ 13N13196. In addition, this work was supported by the German Research Foundation (DFG) and the Technical University of Munich (TUM) in the framework of the Open Access Publishing Fund. The authors would like to thank all contributing project partners, funding agencies, politicians, and stakeholders in different functions in Germany. A very special thanks to the Bavarian Water Authority and Bavarian Environment Agency in Hof for providing us with the quality data to conduct the research. We would also like to thank the language centre of the Technical University of Munich for their consulting in improving the language of the manuscript.

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
