# Peer review of "Hazard maps with differentiated exceedance probability for flood impact assessment"

_Natural Hazards and Earth System Sciences, 2019_

## Referee Comment (RC1) · Anonymous Referee #1 · 28 Jun 2019

The main objective of the paper is to develop a new methodology to generate flood hazard maps. Compared to other studies, the new methodology considers the exceedance probability of multi-model combinations based on forecasted peak discharges derived by a set of hydrological models. In addition, the type of building (purpose & structure) is considered to define the hazard at an object. This is important in terms of prioritization for e.g. early warning and emergency planning.

General comments:

The paper is very interesting and the main objective to create hazard maps or doing impact assessment with a transparent declaration and consideration of uncertainties is desirable. Especially the presented approach using confidence intervals of a hydrological forecast ensemble is interesting and has potential. However, there are doubts

about some major points in this study.

It is questionable whether the approach presented in this study "inherently communicates the underlying uncertainties", as stated in the conclusion. Looking at the final map presented in figure 9a), the meaning of Scenario I, II and III is not directly derivable and the coherence of all classifications and the different combinations of hydrograph scenarios with building types is not easily comprehensible. Even if the names of scenarios in the map would be changed to "high exceedance probability" (=S I), "average exceedance probability" (=S II) and "low exceedance probability" (=S III) – what would already improve the understanding - it's still questionable whether the multi-model combination presented is the right way to deal with uncertainties in hydrological forecasting for flood impact assessment. To combine the exposure for different confidence interval hydrographs in a new scenario defined by the same exceedance probability is not very elegant. According to the output of the ensemble members, the M50% confidence interval hydrograph as used in the paper (= best-model = median) is the hydrograph that is forecasted as the most likely one and therefore, to define scenarios with low, average or high exceedance probabilities based on multi-model combinations referring to different confidence intervals is misleading. I try to exemplify this issue on table 3: The way the M%-hydrographs from the ensemble forecast are used would imply that the higher the confidence interval, the lower the exceedance probability of an event. But this is not the correct way to implement the confidence intervals here. At the time of the forecast and according to the model ensemble, it is as likely that a M25% or a M75% (= 50% confidence interval) event appears (when I understood it right that for example the lower 80% confidence interval figure 2 corresponds to the M10% hydrograph -> 80% confidence interval means that 80% of the forecasted cases will also be in this range and 20% not -> 10% at the lower and another 10% at the upper end of the scale). In general, the terminology used in this study is also misleading, as for example the 50% confidence interval discharge is used as the median (= M50% = best model), whereas the 50% confidence interval would correspond to the M25% discharge and M75% discharge -> 25th and 75th discharge percentile or 0.25 / 0.75 quantile.

I think that two (hydro-)statistical approaches were unconsciously mixed – the one of confidence intervals of a model ensemble forecast and the general probability of discharge to exceed a certain value (extreme value statistics are not any more relevant at the time of the forecast). When applying the method with the confidence intervals correctly, it appears that scenario I (defined as high exceedance probability) is the most unlikely scenario according to the model forecast. Therefore, the multi-model combination is not working as supposed.

In addition, I don't see the a fundamental novelty of this approach compared to the cited study by Zarzar et al. (2018), who already presents the use of a multi-model ensemble framework based on hydrological ensemble members for visualizing flood inundation uncertainty. I agree with the authors that the use of confidence intervals in the case of a high number of hydrological forecast members can support a transparent declaration of forecast uncertainties.

Besides all of this, to my point of view the display of the inundation pathways and extend is necessary for a use in early warning systems, emergency planning or flood impact assessment. The approach of a building hazard map doesn't provide enough details to intervene (where does the water come in contact with the building, where are the "weak spots" in the river- and floodplain-system).

In addition to these major points, there are some further remarks in the specific comments.

Specific comments

Section 2.1, 2.2 + supplement tables: It is mentioned that the particular focus of this study is on the development of the post-processing part (classification and multi-model combination, mapping). Therefore, the two parts of the framework that are already developed and explained similarly in Beg et al. (2018) and Bhola et al. (2018a, 2018b) should be shortened, as for example a detailed explanation of the model LARSIM, FloodEvac and HEC-RAS is not needed to understand the context of this study. Nevertheless, Figure 3 helps to understand the setup of the multi-model approach. It should be better explained, where for example the M10% hydrograph can be found in such a graphic (consistent use of confidence intervals). It would be helpful if you could add the forecast ensemble for the virtual station that was used as upper boundary condition and maybe mark the used hydrographs.

Section 2.3: The main literature cited (Krieger et al., 2017) is specific to Germany and is not peer-reviewed. In which way are the classification methods for buildings and hazard types comparable with international, peer-reviewed literature? E. g. Jonkman et al. (2008), Dutta et al. (2003), as well as Thieken et al. (2008) and Kreibich et al. (2010) on german scale, differentiate loss estimations of residential/industrial/commercial etc. buildings due to different vulnerability, whereas here these types are represented in the same class here (III).

p. 8, lines 17 + 18; figure 4; p. 14, line 12: Check the numbers given on p. 8 – they do obviously not match the numbers per class in figure 4 and also not with the statement on p. 14, that the most buildings were classified in the classes II and III. I'm also doubting the usefulness of building class I, as there are parks included (which are not buildings) and there is only one member. Furthermore, it's questionable if the applied classification method in the context of the study makes sense, as relative to total number of 2695 buildings, creating separate classes with 1 and 20 (based on figure 4, I would assume that this is class IV) buildings respectively, lead to underrepresentation of these classes. Based to that, it's not surprising that for example the low exceedance scenario is similar to the M50% and M75%, as the buildings are mostly distributed in the classes combined with these hydrograph scenarios.

Figure 8 + 9, p. 15 line 1 - 3: In this context, you should maybe explain somewhere, how you assign the hazard for the same building (finally in figure 9) that is in a moderate hazard zone for scenario I but then rise to a very high hazard in scenario III (is the potential in the forecasted event to have very high hazard at a particular building somehow considered?).
p. 16, lines 4 – 11: In this part, the time issue in real-time assessment of the framework is discussed. As well in the introduction it is mentioned, that flood forecasts might be restricted to computational time of the models. Please provide information about the lead- and run-time of the hydrological forecast model, the computational time for the HD model with each hydrograph and, therefore, how much time would be left for authorities to intervene. This is evident for early warning and emergency planning. It would of course also be interesting how the offline maps from Bhola et al. (2018a) would perform compared to the modelled confidence hydrographs.

Technical corrections

In general: As already mentioned, the use of the term confidence interval discharge in combination with the M% HD outputs seems not to be correct. Please replace confidence interval with "percentile" when referring to M%-levels – or change these levels accordingly.

p. 1, line 21: It would maybe help if you would explain a bit more in detail, what in this study is meant as multi-model combination. Based on this first explanation, one would assume that the building classification is not part of the multi-model combination and this consists only of the HD and hydrological model (which, according to figure 1, is not the case). The term is also used in various ways: multi-model, multi-model combination, multi-model ensemble combination, multi-model combination scenarios. E.g. in the first sentence of the discussion section, the context is given only to hydrology. If this is the case, then figure 1 should be adapted. This can be a bit confusing.

p. 8, line 17 + 18: As mentioned, check the numbers for each class and compare with figure and other statements.

p. 11, line 11/12: As mentioned in special comments, to my point of view the confidence intervals cannot be used to describe exceedance probabilities in the way it was done here.

p. 11, line 22: You refer to post-event information that "no serious damage was re-ported" -> p. 14 line 4: "figure 7c is in agreement with the post-event information" -> 7c = M50% discharge scenario. According to figure 6, in this scenario 126 buildings are exposed, 67 classified with high hazard -> How does that fit to "no serious damage was reported"?

Figure 7, 8 and 9 b-d: Legends (and building numbers in 9b) are not readable. Also the maps themselves are at the edge of readability. The red circles in figure 9 are not in the legend and are never mentioned in the text (see comments below)? Is figure 9d really necessary? p. 13, line 12 – 15: This error should be eliminated by either using water depth derived from water surface elevation [m a. s. l. ] minus ground level of building [m a. s. l. ] or by removing the river channel elements from the dataset you used to assign to the buildings. Check out Bermúdez and Zischg (2018).

p. 14, line 17 – 22: Please add the information that you in this specific case compare the red circles from M50% and the multi-model map. ID 1393 is not recognizable in figure 9d. p. 16. Line 3 + 4: What is the computational time of the whole framework? What's the lead-time used here?

Publication bibliography

Bermúdez, María; Zischg, Andreas Paul (2018): Sensitivity of flood loss estimates to building representation and flow depth attribution methods in micro-scale flood mod-elling. In Nat Hazards 92 (3), pp. 1633–1648. DOI: 10.1007/s11069-018-3270-7.

Dutta, Dushmanta; Herath, Srikantha; Musiake, Katumi (2003): A mathematical model for flood loss estimation. In Journal of Hydrology 277 (1-2), pp. 24–49. DOI: 10.1016/S0022-1694(03)00084-2. Jonkman, S. N.; Bočkarjova, M.; Kok, M.; Bernardini, P. (2008): Integrated hydrodynamic and economic modelling of flood damage in the Netherlands. In Ecological Economics 66 (1), pp. 77–90. DOI: 10.1016/j.ecolecon.2007.12.022.

[Figure]

Kreibich, Heidi; Seifert, Isabel; Merz, Bruno; Thieken, Annegret H. (2010): Development of FLEMOcs – a new model for the estimation of flood losses in the commercial sector. In Hydrological Sciences Journal 55 (8), pp. 1302–1314. DOI: 10.1080/02626667.2010.529815.

Thieken, A. H.; Olschewski, A.; Kreibich, H.; Kobsch, S.; Merz, B. (2008): Development and evaluation of FLEMOps – a new F lood L oss E stimation MO del for the p rivate s ector. In D. Proverbs, C. A. Brebbia, E. Penning-Rowsell (Eds.): Flood Recovery, Innovation and Response I. FLOOD RECOVERY, INNOVATION AND RESPONSE 2008. London, England, 02.07.2008 - 03.07.2008. Southampton, UK: WIT Press, pp. 315–324.

---

## Referee Comment (RC2) · Anonymous Referee #2 · 1 Nov 2019

General Comments ——————- Dear authors, thanks for the well designed and inspirational paper. I was asked in October 2018 to review the papger, when another colleague already did a detailed review and gave very valuable comments to the paper. I agree with his comments and underline his implicit proposals for revision. Consequently I will concentrate on additional comments, which might have partial minor importance.

Detailed Comments ——————

Title The title is short (which I prefer) and defines the general topic of the paper. As the core of the paper deals with the flood impact to buildings classified by the building use, I propose to add this to the title e.g. by a more specific definition of "flood impact assessment" in the title.

[Figure]

Page 3 line 12 It is written "The framework to generate building hazard maps in REAL TIME ..." In the paper I did not find any information about this real time feature of the framework. I assume real time did not mean on-time. Can you specify the meaning of real time or simply skip the term ? e.g.: Is real time related to the calculation time of the frame work tools (24 h real time -> 24 calculation time (or less) ?

Application Domain The paper is using as case study a smaller German city with a specific topographic situation and type of river size. Such situations might exist in other parts in Germany, Europe as well as the whole world, but there might be also buildings in other environments (e.g. urban area with "plain" topography, "big" cities/metropolises, large rivers with different flow characteristic). The paper is focusing on the method and not on the case study it might be useful for the reader to have a paragraph about the type of case studies suitable for the application of the proposed method (maybe including the limits of the methods for other types of case studies). It might be also helpful to specify the required data to apply the method for other case studies. In the paper the used data is partial described in different chapters, but esp. the type/quality/level of details of the requested building data for this model should be describes. Ar the four classes in this case study specific for the case study or a general approach ?

Chapter 2.2 shortly describes the basics of the 2D hydrodynamic modeling. I'm missing the description about the handling of the buildings in this model. Are they included explicitly by their shape in the grid and excluded from the flow calculation (no flow through the building) ? Is the urban area including buildings "only" considered by a different (but global) roughness value (Table 2 suppl.)? These two approaches might lead to different water levels at the buildings.

Figure 4 I propose to change the color for Class I, as it is very difficult to distinguish between the light gray of Class I and the lighter gray of the background image. Maybe Class I and II should be not gray but yellow and lighter yellow ocher coming from Red and orange for class IV and III.

(Page 10 line 14) not so important: the comment about the underground metro access is in general correct, but is there a metro in this "small" city ?

Figure 5 The idea of the figure is well chosen, but it took me some time to understand this. Assuming my interpretation is correct, I think this is not a summing up of components (+ operator) which is equal (= operator) to the part on the right side of this equation. It should be not a sum, maybe a selection I propose to replace the = by a ->

The discussion and conclusion is touching in some parts the application aspects for the target users (e.g. page 15). It would be useful to have an explicit discussion about the "progress" of the proposed methods to produce hazard maps for the target users (advantage and disadvantages). With other words: to evaluate the method from the target users point of view and not "only" from the hazard map producer point of view. As the focus of the paper is on the method the application oriented view might be considered at least by some sentences/a paragraph in the conclusions (outlook part).

---

## Author Comment (AC2) · 10 May 2020

General Comments

Dear authors, thanks for the well designed and inspirational paper. I was asked in October 2018 to review the paper, when another colleague already did a detailed review and gave very valuable comments to the paper. I agree with his comments and underline his implicit proposals for revision. Consequently I will concentrate on additional comments, which might have partial minor importance.

Authors:

We sincerely appreciate the constructive feedback from the reviewer in improving the quality of the manuscript. We have addressed all the review comments in the revised manuscript.

After revision of reviews 1, we have changed the term "confidence interval" to percentile and "exceedance probability" to risk perception.

The followings are our point-by-point responses to the reviewer's quotes

Detailed Comments

Reviewer: Title The title is short (which I prefer) and defines the general topic of the paper. As the core of the paper deals with the flood impact to buildings classified by the building use, I propose to add this to the title e.g. by a more specific definition of "flood impact assessment" in the title.

Authors: We have updated the title

> "Buildings hazard maps with differentiated risk perception for flood impact assessment"

Reviewer: Page 3 line 12 It is written "The framework to generate building hazard maps in REAL TIME ..." In the paper I did not find any information about this real time feature of the framework. I assume real time did not mean on-time. Can you specify the meaning of real time or simply skip the term ? e.g.: Is real time related to the calculation time of the frame work tools (24 h real time -> 24 calculation time (or less) ?

Authors: We have omitted the term from methodology. The real-time refers to producing the maps as the discharge are forecasted. We have provided more provided information on the run-time of the entire map production. In Discussion Page 20: Line 12

> "In this study, a 50-member ensemble forecast was used from Beg et al. (2018) where the entire process took 25 minutes in a 3-core desktop in parallel mode to generate a forecast of 12 hours. Various percentile discharges were then run simultaneously in the HD model, which required 30 min to simulate a 12-hour event on an 8-core, 2.4 GHz (Intel E5-2665), including the initial start (Bhola et al. 2018a). Post-processing of the model results would consume an additional 15 min. Therefore, real-time hazard maps are delivered to decision makers in 70 min."

Reviewer: Application Domain The paper is using as case study a smaller German city with a specific topographic situation and type of river size. Such situations might exist in other parts in Germany, Europe as well as the whole world, but there might be also buildings in other environments (e.g. urban area with "plain" topography, "big" cities/metropolises, large rivers

with different flow characteristic). The paper is focusing on the method and not on the case study it might be useful for the reader to have a paragraph about the type of case studies suitable for the application of the proposed method (maybe including the limits of the methods for other types of case studies). It might be also helpful to specify the required data to apply the method for other case studies. In the paper the used data is partial described in different chapters, but esp. the type/quality/level of details of the requested building data for this model should be describes. Are the four classes in this case study specific for the case study or a general approach ?

Authors: We added references to other classification in literature.

In methodology, Page 8: Line 6-10

"There are various classifications of land use features available in literature. Dutta et al. (2003) have used direct and indirect damages as the basis of their classification and classified their study area in residential and non-residential categories. Jonkman et al. (2008) have classified urban features in residential, businesses, commercial and public property and agricultural to estimate flood loss. Furthermore, vulnerability was the basis of classification in residential (Thieken et al., 2008) and Industrial & commercial sectors (Kreibich et al., 2010) in order to estimate flood losses. "

And we justify the one we used based on: Page 8: Line 10-13

"We have used damage potential of a building as a basis for classification in order to focus on the flood impact assessment. Building damage potential is required for a variety of flood mitigation planning activities including flood damage assessment, multi-hazard analyses and emergency measures (Shultz, 2017)."

And we added also in Page 8: Line 16-18

"In the authors opinion by keeping our classification simple will likely fit a vast majority of cities regardless of their size. In any case we acknowledge that the number of classes/ criteria can be changed/adapted depending on the aim of the forecast."

For applicability in other study areas, In discussion, Page 20:Line 10-11

"Overall, the methodology is independent of the choice of models, i.e. hydrological and HD, and is transferable to other study areas."

And In conclusion, Page Page 21: Line 25-28

"Further research investigating multi-model combinations and validation in other study areas may be beneficial. A more extensive study on the validation of the multi-model combination may be required, possibly by using measuring gauges, post-event survey (as conducted in Thieken et al., 2005), satellite images (as in Triglav-Čekada and Radovan, 2013), and/or crowd-sourced data (Bhola et al., 2018b)."

Reviewer: Chapter 2.2 shortly describes the basics of the 2D hydrodynamic modeling. I'm missing the description about the handling of the buildings in this model. Are they included explicitly by their shape in the grid and excluded from the flow calculation (no flow through the building) ? Is the urban area including buildings "only" considered by a different (but global) roughness value (Table 2 suppl.)? These two approaches might lead to different water levels at the buildings.

Authors: We have added the information on handling the buildings and assigning hazard. In 2.2 Hydrodynamic modelling Page 7: Line21-23

> "The buildings are explicitly included using their shape in the mesh and are excluded from the flow calculation by assigning a high roughness value. To assign hazard to a building, the maximum water depth of all the neighboring cells was used."

Reviewer: Figure 4 I propose to change the color for Class I, as it is very difficult to distinguish between the light gray of Class I and the lighter gray of the background image. Maybe Class I and II should be not gray but yellow and lighter yellow ocher coming from Red and orange for class IV and III.

Authors: Thank you for pointing it out, we have changed the color so they are visible.

[Figure]

**Figure 1. City of Kulmbach and building use classification. (Data source: Bavarian Ministry of the Interior, for Building and Transport and Water Management Authority Hof).**

Reviewer: (Page 10 line 14) not so important: the comment about the underground metro access is in general correct, but is there a metro in this "small" city ?

Authors: The classification is general and can be applied in any other city. In this study area there is no metro but there are class IV buildings.

Reviewer: Figure 5 The idea of the figure is well chosen, but it took me some time to understand this. Assuming my interpretation is correct, I think this is not a summing up of components (+ operator) which is equal (= operator) to the part on the right side of this equation. It should be not a sum, maybe a selection I propose to replace the = by a ->

Authors: We have replaces = by ->

[Figure]

**Figure 2. An example of a multi-model combination in which the four building classes I, II, III and IV are assigned to the 2D HD model results of 25%, 50%, 75% and 90% respectively.**

Reviewer: The discussion and conclusion is touching in some parts the application aspects for the target users (e.g. page 15). It would be useful to have an explicit discussion about the "progress" of the proposed methods to produce hazard maps for the target users (advantage and disadvantages). With other words: to evaluate the method from the target users point of view and not "only" from the hazard map producer point of view. As the focus of the paper is on the method the application oriented view might be considered at least by some sentences/a paragraph in the conclusions (outlook part).

Authors: Thank you for the comment. The following paragraph was added/changed in the discussion, Page 21: Line 5-10.

> "Therefore, our methodology would allow the target users to benefit from hazard maps enabling them to better prioritise and coordinate evacuation planning based on the highest forecasted impacts. The maps could further serve as a tool for flood risk assessment. The methodology can be used for flood mitigation and flood forecast planning in the form of emergency management training, where forecasted hazard scenarios can be presented to the training groups. By visualising inundation scenarios, potential damage at the building's level which have been prioritized based on a desired classification, can be estimated with this methodology and made available together with each forecasted scenario."

And in outlook, Page 21: Line 29-Page 22 Line 5

> "In future, damage potential classification can further be improved by including additional criteria, such as population density or water quality, and with it extend the applicability of this method. For example, the assessment of the damage potential of commercial enterprises, substances or machinery containing elements that which could be a source of water pollution could be included (Krieger et al., 2017). In addition, other classification methods for buildings and hazard types should be evaluated, especially to further dissect the impact of class III in commercial and industrial. Finally, the output of the framework could be extended to hazard maps uploaded in a web-based GIS system to improve visualization, along with providing layers of additional information, such as inundation pathways and hot spot to aid in planning. The latter would enhance the usefulness to different target users, such as planners, decision makers and flood forecasting agencies."

---

## Author Response (AR1)

The main objective of the paper is to develop a new methodology to generate flood hazard maps. Compared to other studies, the new methodology considers the exceedance probability of multi-model combinations based on forecasted peak discharges derived by a set of hydrological models. In addition, the type of building (purpose & structure) is considered to define the hazard at an object. This is important in terms of prioritization for e.g. early warning and emergency planning.

Authors: We do sincerely appreciate the detailed and very good comments from the reviewers', and we have acknowledged their contribution in improving the quality of this paper. We hope that we have now addressed all the reviewers' comments satisfactorily in the revised manuscript.

After recommendation from other reviewer, we have updated the title

"Buildings hazard maps with differentiated risk perception for flood impact assessment"

Please find our point-by-point responses to the comments

Reviewer: General comments:

- The paper is very interesting and the main objective to create hazard maps or doing impact assessment with a transparent declaration and consideration of uncertainties is desirable. Especially the presented approach using confidence intervals of a hydrological forecast ensemble is interesting and has potential. However, there are doubts about some major points in this study.
- It is questionable whether the approach presented in this study "inherently communicates the underlying uncertainties", as stated in the conclusion. Looking at the final map presented in figure 9a), the meaning of Scenario I, II and III is not directly derivable and the coherence of all classifications and the different combinations of hydrograph scenarios with building types is not easily comprehensible. Even if the names of scenarios in the map would be changed to "high exceedance probability" (=S I), "average exceedance probability" (=S II) and "low exceedance probability" (=S III) – what would already improve the understanding - it's still questionable whether the multi-model combination presented is the right way to deal with uncertainties in hydrological forecasting for flood impact assessment. To combine the exposure for different confidence interval hydrographs in a new scenario defined by the same exceedance probability is not very elegant. According to the output of the ensemble members, the M50% confidence interval hydrograph as used in the paper (= best-model = median) is the hydrograph that is forecasted as the most likely one and therefore, to define scenarios with low, average or high exceedance probabilities based on multi-model combinations referring to different confidence intervals is misleading. I try to exemplify this issue on table 3: The way the M%-hydrographs from the ensemble forecast are used would imply that the higher the confidence interval, the lower the exceedance probability of an event. But this is not the correct way to implement the confidence intervals here. At the time of the forecast and according to the model ensemble, it is as likely that a M25% or a M75% (= 50% confidence interval) event appears (when I

understood it right that for example the lower 80% confidence interval figure 2 corresponds to the M10% hydrograph -> 80% confidence interval means that 80% of the forecasted cases will also be in this range and 20% not -> 10% at the lower and another 10% at the upper end of the scale). In general, the terminology used in this study is also misleading, as for example the 50% confidence interval discharge is used as the median (= M50% = best model), whereas the 50% confidence interval would correspond to the M25% discharge and M75% discharge -> 25th and 75th discharge percentile or 0.25 / 0.75 quantile.

I think that two (hydro-)statistical approaches were unconsciously mixed – the one of confidence intervals of a model ensemble forecast and the general probability of discharge to exceed a certain value (extreme value statistics are not any more relevant at the time of the forecast). When applying the method with the confidence intervals correctly, it appears that scenario I (defined as high exceedance probability) is the most unlikely scenario according to the model forecast. Therefore, the multi-model combination is not working as supposed.

Authors: Thank you for pointing out the error, as suggested we have replaced the term percentile in the revised manuscript and corrected as in Figure 1 (Figure 3 in Manuscript). We also agree that the exceedance probability is not the right choice of word to describe the scenarios, hence we have replaced it by "risk perception" and have added the scenario description in Figure 2 (Figure 9a in manuscript).

[Figure]

    **(a) Ködnitz**                           **(b) Kauerndorf**

**Figure 1. Hindcasted flood event of January 2011: measured discharge hydrograph along with 95%, 90% and 75% percentile discharges for gauges a) Ködnitz and b) Kauerndorf (Data based on Beg et al., 2018).**

[Figure]

**Figure 2: Multi-model combination**

In addition, we stated that these maps offer an alternative way to communicate the underlying uncertainties. It will be interesting for authors to measure how the scientific community and practitioners will receive the methodology. In conclusions, Page 22: Line 21-24

> "In summary, we have presented a new methodology for flood impact assessment using a multi-model combination in the form of *building hazard maps with differentiated risk perceptions.* These maps offer an alternative way to communicate the underlying uncertainties in forecasting models and are ready-to-use for decision-makers in the field of flood risk management.".

- In addition, I don't see the a fundamental novelty of this approach compared to the cited study by Zarzar et al. (2018), who already presents the use of a multi-model ensemble framework based on hydrological ensemble members for visualizing flood inundation uncertainty. I agree with the authors that the use of confidence intervals in the case of a high number of hydrological forecast members can support a transparent declaration of forecast uncertainties.

Authors: We have now differentiates the work done by Zarzar et al. (2018), they have used an average of HD model raster outputs to obtain the percentage of ensemble agreement. Whereas, we have exported HD models on buildings as urban features and proposed a new methodology of multi-model combination. In Introduction, Page 3: Line 4-7

> "Zarzar et al. (2018) have used a multi-model combination framework consisting of hydro-metrological and HD models to visualise flood inundation uncertainties in which they have

used an average of HD model raster outputs to obtain the percentage of ensemble agreement."

- Besides all of this, to my point of view the display of the inundation pathways and extend is necessary for a use in early warning systems, emergency planning or flood impact assessment. The approach of a building hazard map doesn't provide enough details to intervene (where does the water come in contact with the building, where are the "weak spots" in the river- and floodplain-system).

Authors: We acknowledge the importance of the inundation extent; we have proposed to use hazards maps in addition to other layers of information, such as inundation map for planning. In conclusion, Page 23-Line 12-16

"Finally, the output of the framework can be extended to hazard maps uploaded in a web-based GIS system to improve visualization, along with providing layers of additional information, such as inundation pathways and weak spots in the river and floodplains to provide sufficient details to intervene (aid in planning). This additional information would enhance the usefulness to different target users, such as planners, decision-makers and flood forecasting agencies."

Reviewer: In addition to these major points, there are some further remarks in the specific comments.

Specific comments

- Section 2.1, 2.2 + supplement tables: It is mentioned that the particular focus of this study is on the development of the post-processing part (classification and multi-model combination, mapping). Therefore, the two parts of the framework that are already developed and explained similarly in Beg et al. (2018) and Bhola et al. (2018a, 2018b) should be shortened, as for example a detailed explanation of the model LARSIM, FloodEvac and HEC-RAS is not needed to understand the context of this study. Nevertheless, Figure 3 helps to understand the setup of the multi-model approach. It should be better explained, where for example the M10% hydrograph can be found in such a graphic (consistent use of confidence intervals). It would be helpful if you could add the forecast ensemble for the virtual station that was used as upper boundary condition and maybe mark the used hydrographs.

Authors: We agree with the reviewer comment regarding shortening the details, however, additional details are provided in order to reproduce the methodology, as suggested by the editor.

In the revised manuscript, we have added data labels in Figure 1 to clearly present the percentiles. These two stations where used as the input boundary conditions upstream of the model. The virtual gauge is nothing but the addition of these two hydrographs, we present here the data at gauges upstream of the city.

- Section 2.3: The main literature cited (Krieger et al., 2017) is specific to Germany and is not peer-reviewed. In which way are the classification methods for buildings and hazard types comparable with international, peer-reviewed literature? E. g. Jonkman et al. (2008), Dutta et al. (2003), as well as Thieken et al. (2008) and Kreibich et al.

(2010) on german scale, differentiate loss estimations of residential/industrial/commercial etc. buildings due to different vulnerability, whereas here these types are represented in the same class here (III).

Authors: Thank you for providing the studies, we have compared their classification to ours in the revised manuscript. We also stated further break down of class III in conclusion. In Section 2.3.1, Page 8: Line 8-18

"There are various classifications of land use features available in the literature. Dutta et al. (2003) have used direct and indirect damages as the basis of their classification and classified their study area in residential and non-residential categories. Jonkman et al. (2008) have classified urban features in residential, businesses, commercial and public property and agricultural to estimate flood loss. Furthermore, the vulnerability was the basis of classification in residential (Thieken et al., 2008) and industrial and commercial sectors (Kreibich et al., 2010) in order to estimate flood losses. We have used the damage potential of a building as a basis for classification in order to focus on the flood impact assessment. Building damage potential is required for a variety of flood mitigation planning activities including flood damage assessment, multi-hazard analyses and emergency measures (Shultz, 2017). The buildings were classified into four classes based on their function following the recommendation of the German standard for risk management in urban areas in the case of flash floods (Krieger et al., 2017)"

And, in Conclusion, Page 23: Line 11-12

"In addition, other classification methods for buildings and hazard types should be evaluated, especially to further dissect the impact of class III in commercial and industrial."

- p. 8, lines 17 + 18; figure 4; p. 14, line 12: Check the numbers given on p. 8 – they do obviously not match the numbers per class in figure 4 and also not with the statement on p. 14, that the most buildings were classified in the classes II and III. I'm also doubting the usefulness of building class I, as there are parks included (which are not buildings) and there is only one member. Furthermore, it's questionable if the applied classification method in the context of the study makes sense, as relative to total number of 2695 buildings, creating separate classes with 1 and 20 (based on figure 4, I would assume that this is class IV) buildings respectively, lead to underrepresentation of these classes. Based to that, it's not surprising that for example the low exceedance scenario is similar to the M50% and M75%, as the buildings are mostly distributed in the classes combined with these hydrograph scenarios.

Authors: Thank you for pointing out the error, we have corrected in the revised manuscript. The building class I belongs to buildings that are in open green area such as small park and garden building. In Section 2.3.1, Page 8: Line 29-30

"There are a total of 2695 buildings in Figure of which 1, 958, 1716 and 20 buildings were classified in classes I, II, III and IV respectively."

Regarding the building classification, Page 8: Line 15-18

> "The buildings were classified into four classes based on their function following the recommendation of the German standard for risk management in urban areas in the case of flash floods (Krieger et al., 2017)."

We acknowledge that depending on the aim the classification can be adjusted. Page 8: Line 21-22

> "In any case, we acknowledge that the number of classes or criteria can be changed/adapted depending on the aim of the forecast."

We have also added in Page 8: Line 31-32

> "The nature of the data in this case study leads to a differentiated representation of the classes. It should be noted that the classification aims at creating classes based on damage potential, and not on generating clusters with similar sizes."

- Figure 8 + 9, p. 15 line 1 - 3: In this context, you should maybe explain somewhere, how you assign the hazard for the same building (finally in figure 9) that is in a moderate hazard zone for scenario I but then rise to a very high hazard in scenario III (is the potential in the forecasted event to have very high hazard at a particular building somehow considered?).

Authors: Thank you for your comment. This was done based on Table 1, in Results Section 3.2, Page 16: Line 4-11

> "The main objective of the combination is to differentiate the impact of water depths on building classes. Therefore, to design the combinations, a high percentile was assigned to the buildings with a high damage potential class. Each scenario presents a given risk perception of a decision-maker that can be easily adjusted depending on the perception of different stakeholders. Hence, different risk perceptions will lead to different exceedance probability maps. A risk-averse person will likely select low exceedance probabilities for the hazard maps, whereas a risk-seeking person will likely develop risk management strategies based on higher exceedance probabilities. The hazard maps for the three scenarios are shown in **Fehler! Verweisquelle konnte nicht gefunden werden.**."

**Table 1. Scenarios of multi-model combinations based on risk perception.**

| Scenario | Risk perception | Building class | | | |
|---|---|---|---|---|---|
| | | I | II | III | IV |
| I | Risk-seeking | $M_{10\%}$ | $M_{10\%}$ | $M_{25\%}$ | $M_{50\%}$ |
| II | Risk-neutral | $M_{10\%}$ | $M_{25\%}$ | $M_{50\%}$ | $M_{75\%}$ |
| III | Risk-averse | $M_{25\%}$ | $M_{50\%}$ | $M_{75\%}$ | $M_{90\%}$ |

- "p. 16, lines 4 – 11: In this part, the time issue in real-time assessment of the framework is discussed. As well in the introduction it is mentioned, that flood forecasts might be restricted to computational time of the models. Please provide information about the lead- and run-time of the hydrological forecast model, the computational time for the HD model with each hydrograph and, therefore, how much time would be left for

authorities to intervene. This is evident for early warning and emergency planning. It would of course also be interesting how the offline maps from Bhola et al. (2018a) would perform compared to the modelled confidence hydrographs.

Authors: We have provided information on the run-time of the entire map production. In Discussion, Page 22: Line 3-7

"In this study, a 50-member ensemble forecast was used from Beg et al. (2018) where the entire process took 25 minutes in a 3-core desktop in parallel mode to generate a forecast of 12 hours. Various percentile discharges were then run simultaneously in the HD model, which required 30 minutes to simulate a 12-hour event on an 8-core, 2.4 GHz (Intel E5-2665), including the initial start (Bhola et al. 2018a). Post-processing of the model results would consume an additional 15 min. Therefore, real-time hazard maps are delivered to decision-makers in 70 min.

Technical corrections

- In general: As already mentioned, the use of the term confidence interval discharge in combination with the M% HD outputs seems not to be correct. Please replace confidence interval with "percentile" when referring to M%-levels – or change these levels accordingly.

Authors: The term is changed to percentile in the revised manuscript.

- p. 1, line 21: It would maybe help if you would explain a bit more in detail, what in this study is meant as multi-model combination. Based on this first explanation, one would assume that the building classification is not part of the multi-model combination and this consists only of the HD and hydrological model (which, according to figure 1, is not the case). The term is also used in various ways: multi-model, multi-model combination, multi-model ensemble combination, multi-model combination scenarios. E.g. in the first sentence of the discussion section, the context is given only to hydrology. If this is the case, then figure 1 should be adapted. This can be a bit confusing.

Authors: We have used multi-model combination consistently in the revised manuscript. The term is changed to percentile in the revised manuscript. Building classification is part of the entire framework but independent of multi-model combinations. We have focused this study on the multi-model combination. In Methodology, Page 3: Line 23-24.

"The particular focus of this study is on the development of the framework of a multi-model combination in the post-processing component"

- p. 8, line 17 + 18: As mentioned, check the numbers for each class and compare with figure and other statements.

Authors: Thank you for pointing it out, we confirm that the numbers provided are correct in revised manuscript.

- p. 11, line 11/12: As mentioned in special comments, to my point of view the confidence intervals cannot be used to describe exceedance probabilities in the way it was done here.

Authors: We have corrected the term and used percentile consistently.

- p. 11, line 22: You refer to post-event information that "no serious damage was reported" -> p. 14 line 4: "figure 7c is in agreement with the post-event information" -> 7c = M50% discharge scenario. According to figure 6, in this scenario 126 buildings are exposed, 67 classified with high hazard -> How does that fit to "no serious damage was reported"?

Authors: This is explain in the discussion. The high hazard at $M_{50\%}$ is due to their proximity to the Mühl canal. In Discussion, Page 18: Line 8-14

"It should be noted that few buildings show *very high* hazards due to their proximity to the Mühl canal (Figure7**Fehler! Verweisquelle konnte nicht gefunden werden.**a). Even though there was no over-topping of water from the canal, because of buildings geolocation being near to the canal, these were assigned automatically with the highest hazard, starting with a discharge of $M_{10\%}$. Ideally, this should be prevented by removing the river channel elements from the dataset before to assign the water depths to the buildings as in Bermúdez and Zischg (2018). However, and without retract to our conclusions, it was decided not to include it in this work in order to keep the automation process simple."

- Figure 7, 8 and 9 b-d: Legends (and building numbers in 9b) are not readable. Also the maps themselves are at the edge of readability. The red circles in figure 9 are not in the legend and are never mentioned in the text (see comments below)? Is figure 9d really necessary?

Authors: The figures have been resized so the legends are readable, in addition all the figures are provided to the editorial so the final print will be of high-quality and readable. We have omitted figure 9d from the revised manuscript. The red circles are defined in the figure caption as well as in the text.

- p. 13, line 12 – 15: This error should be eliminated by either using water depth derived from water surface elevation [m a. s. l. ] minus ground level of building [m a. s. l. ] or by removing the river channel elements from the dataset you used to assign to the buildings. Check out Bermúdez and Zischg (2018).

Authors: Thank you for your comment. We have added this limitation of our study to the article. In Discussion, Page 18: Line 11-14.

"Ideally, this should be prevented by removing the river channel elements from the dataset before to assign the water depths to the buildings as in Bermúdez and Zischg (2018). However, and without retract to our conclusions, it was decided not to include it in this work in order to keep the automation process simple."

- p. 14, line 17 – 22: Please add the information that you in this specific case compare the red circles from M50% and the multi-model map. ID 1393 is not recognizable in figure 9d. p. 16. Line 3 + 4: What is the computational time of the whole framework? What's the lead-time used here?

Authors: We have added the information and stated the computational time of whole framework. Please see in specific comments.

Publication bibliography

Thank you for providing further literature in helping the quality of te paper, we have added the recommended citations

- Bermúdez, María; Zischg, Andreas Paul (2018): Sensitivity of flood loss estimates to building representation and flow depth attribution methods in micro-scale flood modelling. In Nat Hazards 92 (3), pp. 1633–1648. DOI: 10.1007/s11069-018-3270-7.
- Dutta, Dushmanta; Herath, Srikantha; Musiake, Katumi (2003): A mathematical model for flood loss estimation. In Journal of Hydrology 277 (1-2), pp. 24–49. DOI: 10.1016/S0022-1694(03)00084-2.
- Jonkman, S. N.; Bockarjova, M.; Kok, M.; ˇBernardini, P. (2008): Integrated hydrodynamic and economic modelling of flood damage in the Netherlands. In Ecological Economics 66 (1), pp. 77–90. DOI: 10.1016/j.ecolecon.2007.12.022.
- Kreibich, Heidi; Seifert, Isabel; Merz, Bruno; Thieken, Annegret H. (2010): Development of FLEMOcs – a new model for the estimation of flood losses in the commercial sector. In Hydrological Sciences Journal 55 (8), pp. 1302–1314. DOI: 10.1080/02626667.2010.529815.
- Thieken, A. H.; Olschewski, A.; Kreibich, H.; Kobsch, S.; Merz, B. (2008): Development and evaluation of FLEMOps – a new F lood L oss E stimation MO del for the private s ector. In D. Proverbs, C. A. Brebbia, E. Penning-Rowsell (Eds.): Flood Recovery, Innovation and Response I. FLOOD RECOVERY, INNOVATION AND RESPONSE 2008. London, England, 02.07.2008 - 03.07.2008. Southampton, UK: WIT Press, pp.315–324.

Anonymous Referee #2

General Comments

Dear authors, thanks for the well designed and inspirational paper. I was asked in October 2018 to review the paper, when another colleague already did a detailed review and gave very valuable comments to the paper. I agree with his comments and underline his implicit proposals for revision. Consequently I will concentrate on additional comments, which might have partial minor importance.

Authors:

We appreciate the constructive feedback from the reviewer in improving the quality of the manuscript. We have addressed all the review comments in the revised manuscript.

After revision of reviews 1, we have changed the term "confidence interval" to percentile and "exceedance probability" to risk perception.

The followings are our point-by-point responses to the reviewer's quotes

Detailed Comments

Reviewer: Title The title is short (which I prefer) and defines the general topic of the paper. As the core of the paper deals with the flood impact to buildings classified by the building use, I propose to add this to the title e.g. by a more specific definition of "flood impact assessment" in the title.

Authors: We have updated the title

> "Buildings hazard maps with differentiated risk perception for flood impact assessment"

Reviewer: Page 3 line 12 It is written "The framework to generate building hazard maps in REAL TIME ..." In the paper I did not find any information about this real time feature of the framework. I assume real time did not mean on-time. Can you specify the meaning of real time or simply skip the term ? e.g.: Is real time related to the calculation time of the frame work tools (24 h real time -> 24 calculation time (or less) ?

Authors: We have omitted the term from methodology. The real-time refers to producing the maps as the discharge are forecasted. We have provided more provided information on the run-time of the entire map production. We have provided information on the run-time of the entire map production. In Discussion, Page 22: Line 3-7.

> "In this study, a 50-member ensemble forecast was used from Beg et al. (2018) where the entire process took 25 minutes in a 3-core desktop in parallel mode to generate a forecast of 12 hours. Various percentile discharges were then run simultaneously in the HD model, which required 30 minutes to simulate a 12-hour event on an 8-core, 2.4 GHz (Intel E5-2665), including the initial start (Bhola et al. 2018a). Post-processing of the model results would consume an additional 15 min. Therefore, real-time hazard maps are delivered to decision-makers in 70 min."

Reviewer: Application Domain The paper is using as case study a smaller German city with a specific topographic situation and type of river size. Such situations might exist in other parts in Germany, Europe as well as the whole world, but there might be also buildings in other environments (e.g. urban area with "plain" topography, "big" cities/metropolises, large rivers with different flow characteristic). The paper is focusing on the method and not on the case study it might be useful for the reader to have a paragraph about the type of case studies suitable for the application of the proposed method (maybe including the limits of the methods for other types of case studies). It might be also helpful to specify the required data to apply the method for other case studies. In the paper the used data is partial described in different chapters, but esp. the type/quality/level of details of the requested building data for this model should be describes. Are the four classes in this case study specific for the case study or a general approach ?

Authors: We added references to other classification in literature.

In methodology section 2.3.1, Page 8: Line 8-18

> "There are various classifications of land use features available in the literature. Dutta et al. (2003) have used direct and indirect damages as the basis of their classification and classified their study area in residential and non-residential categories. Jonkman et al. (2008) have classified urban features in residential, businesses, commercial and public property and agricultural to estimate flood loss. Furthermore, the vulnerability was the basis of classification in residential (Thieken et al., 2008) and industrial and commercial sectors (Kreibich et al., 2010) in order to estimate flood losses. We have used the damage potential of a building as a basis for classification in order to focus on the flood impact assessment. Building damage potential is required for a variety of flood mitigation planning activities including flood damage assessment, multi-hazard analyses and emergency measures (Shultz, 2017). The buildings were classified into four classes based on their function following the recommendation of the German standard for risk management in urban areas in the case of flash floods (Krieger et al., 2017)."

And we added also in Page 8: Line 20-22

> "In the authors' opinion by keeping our classification simple will likely fit a vast majority of cities regardless of their size. In any case, we acknowledge that the number of classes or criteria can be changed/adapted depending on the aim of the forecast."

For applicability in other study areas, In discussion, Page 22: Line 1-2

> "Overall, the methodology is independent of the choice of models, i.e. hydrological and HD, and is transferable to other study areas."

And In conclusion, Page 23: Line 4-8

> "Further research investigating multi-model combinations and validation in other study areas may be beneficial. A more extensive study on the validation of the multi-model combination may be required, possibly by using measuring gauges, post-event survey (as conducted in Thieken et al., 2005), satellite images (as in Triglav-Čekada and Radovan, 2013), and/or crowd-sourced data (Bhola et al., 2018b)."

Reviewer: Chapter 2.2 shortly describes the basics of the 2D hydrodynamic modeling. I'm missing the description about the handling of the buildings in this model. Are they included explicitly by their shape in the grid and excluded from the flow calculation (no flow through the building) ? Is the urban area including buildings "only" considered by a different (but global) roughness value (Table 2 suppl.)? These two approaches might lead to different water levels at the buildings.

Authors: We have added the information on handling the buildings and assigning hazard. In 2.2 Hydrodynamic modelling Page 7: Line21-23

> "The buildings are explicitly included using their shape in the mesh and are excluded from the flow calculation by assigning a high roughness value. To assign hazard to a building, the maximum water depth of all the neighbouring cells was used."

Reviewer: Figure 4 I propose to change the color for Class I, as it is very difficult to distinguish between the light gray of Class I and the lighter gray of the background image. Maybe Class I and II should be not gray but yellow and lighter yellow ocher coming from Red and orange for class IV and III.

Authors: Thank you for pointing it out, we have changed the color so they are visible in Figure 3.

[Figure]

**Figure 3. City of Kulmbach and building use classification. (Data source: Bavarian Ministry of the Interior, for Building and Transport and Water Management Authority Hof).**

Reviewer: (Page 10 line 14) not so important: the comment about the underground metro access is in general correct, but is there a metro in this "small" city ?

Authors: The classification is general and can be applied in any other city. In this study area there is no metro but there are class IV buildings.

Reviewer: Figure 5 The idea of the figure is well chosen, but it took me some time to understand this. Assuming my interpretation is correct, I think this is not a summing up of components (+ operator) which is equal (= operator) to the part on the right side of this equation. It should be not a sum, maybe a selection I propose to replace the = by a ->

Authors: We have replaces = by -> in Figure 4

[Figure]

**Figure 4: An example of a multi-model combination in which the four building classes I, II, III and IV are assigned to the 2D HD model results of 25%, 50%, 75% and 90% respectively.**

Reviewer: The discussion and conclusion is touching in some parts the application aspects for the target users (e.g. page 15). It would be useful to have an explicit discussion about the "progress" of the proposed methods to produce hazard maps for the target users (advantage and disadvantages). With other words: to evaluate the method from the target users point of view and not "only" from the hazard map producer point of view. As the focus of the paper is on the method the application oriented view might be considered at least by some sentences/a paragraph in the conclusions (outlook part).

Authors: Thank you for the comment. The following paragraph was added/changed in the discussion, Page 22: Line 13-19.

[revised manuscript text omitted]

[Figure]

**(aa) M₁₀%**

[Figure]

**(bb) M₂₅%**

[Figure]

**(ce) M**$_{50\%}$

**(a) M**$_{10\%}$

[Figure]

**(d) M₇₅%**

**(ee) M₉₀%**

**Figure 7. Flood inundation and building hazard maps for five percentiles discharge hydrographs.**

**3.2 Multi-model combination**

Three combination scenarios based on risk-seeking, risk-neutral and risk-averse exceedance probability approach were  designed to illustrate the methodology developed in this  study and  are presented in Table 3. The main objective of the combination is to differentiate the impact of water depths on
5  building classes. Therefore, to design the combinations, a high percentile was assigned to the buildings with high damage potential class. Each scenario presents a given risk perception of a decision-maker that can be easily adjusted depending on the perception of different stakeholders. Hence, different risk perceptions will lead to different exceedance probability maps. A  risk-averse person will likely select low exceedance probability for the hazard maps, whereas a  risk-seeking person will likely
10  develop risk management strategies based on higher exceedance probabilities . The hazard maps for the three scenarios are shown in Figure 8.

**Table 3.** Scenarios of multi-model combination based on risk perception.

| Scenario | Risk perception | Building class | | | |
|:---:|:---:|:---:|:---:|:---:|:---:|
| | | I | II | III | IV |
| **I** | Risk-seeking | $M_{10\%}$ | $M_{10\%}$ | $M_{25\%}$ | $M_{50\%}$ |
| **II** | Risk-neutral | $M_{10\%}$ | $M_{25\%}$ | $M_{50\%}$ | $M_{75\%}$ |
| **III** | Risk-averse | $M_{25\%}$ | $M_{50\%}$ | $M_{75\%}$ | $M_{90\%}$ |

[Figure]

**(a) Scenario I: risk-seeking**

[Figure]

**(bb) Scenario II: risk-neutral**

(c) Scenario III: risk-averse,

[revised manuscript text omitted]

Thieken, A. H., Bessel, T., Kienzler, S., Kreibich, H., Müller, M., Pisi, S., and Schröter, K.: The flood of June 2013 in Germany: how much do we know about its impacts?, Nat. Hazards Earth Syst. Sci., 16, 1519-1540, doi: 10.5194/nhess-16-1519-2016, 2016.

Todini, E.: Flood Forecasting and Decision Making in the new Millennium. Where are We?, Water Res. Manag., 31, 3111-3129, doi: 10.1007/s11269-017-1693-7, 2017.

Triglav-Čekada, M., and Radovan, D.: Using volunteered geographical information to map the November 2012 floods in Slovenia, Nat. Hazards Earth Syst. Sci., 13, 2753-2762, doi: 10.5194/nhess-13-2753-2013, 2013.

Uusitalo, L., Lehikoinen, A., Helle, I., and Myrberg, K.: An overview of methods to evaluate uncertainty of deterministic models in decision support, Environ. Modell. Softw., 63, 24-31, doi: 10.1016/j.envsoft.2014.09.017, 2015.

Voigt, S., Kemper, T., Riedlinger, T., Kiefl, R., Scholte, K., and Mehl, H.: Satellite Image Analysis for Disaster and Crisis-Management Support, IEEE T. Geosci. Remote., 45, 1520-1528, doi: 10.1109/TGRS.2007.895830, 2007.

Wasserwirtschaftsamt Hof: Gebiet des Mains: https://www.wwa-ho.bayern.de/hochwasser/hochwasserereignisse/januar2011/main/index.htm, access: 27.03.2019, 2011.

Weigel, A. P., Liniger, M. A., and Appenzeller, C.: Can multi-model combination really enhance the prediction skill of probabilistic ensemble forecasts?, Q. J. Roy. Meteor. Soc., 134, 241-260, doi: 10.1002/qj.210, 2008.

Zarzar, C. M., Hosseiny, H., Siddique, R., Gomez, M., Smith, V., Mejia, A., and Dyer, J.: A Hydraulic MultiModel Ensemble Framework for Visualizing Flood Inundation Uncertainty, J. Am. Water Resour. As., 54, 807-819, doi: 10.1111/1752-1688.12656, 2018.

Zheng, X., Tarboton, D. G., Maidment, D. R., Liu, Y. Y., and Passalacqua, P.: River Channel Geometry and Rating Curve Estimation Using Height above the Nearest Drainage, J. Am. Water Resour. As., 54, 785-806, doi: 10.1111/1752-1688.12661, 2018.

Jonkman, S. N.; Bockarjova, M.; Kok, M.; ˇBernardini, P. (2008): Integrated hydrodynamic and economic modelling of flood damage in the Netherlands. In Ecological Economics 66 (1), pp. 77–90. DOI: 10.1016/j.ecolecon.2007.12.022.Kreibich, Heidi; Seifert, Isabel; Merz, Bruno;

---

## Referee Report (RR1)

I would like to thank the authors for giving answer to all my concerns and for implementing many of the suggested corrections. In this new version, it is much clearer with which hydrological input the hydraulic models were run and there is now much less confusion about this part.

However, in his comments to the resubmission, the editor doubted the suitability of 'risk perception' used as replacement for 'exceedance probabilty'. I share his doubts. The paper cited with context to risk perception (Paulsen et al., 2012) is maybe not the best choice, as this is about "the development of economic risk preference from childhood to adulthood" with a very economic focus and I would say that in flood risk management stakeholders or decision-makers would maybe not select a map that is described as 'risk-seeking' to initiate mitigation measures. From the view of a decision-maker, I should not have to define if I am risk-seeking, -neutral or –averse to select a suitable hazard map for initiating mitigation measures, instead I should receive maps that show what can potentially happen in an objective way. Of course the selected strategy then depends on the individual risk perception. There are other, to my opinion more suitable papers about risk perception with the context to flood risk management (see e.g. Botzen et al. 2009). The authors should have a closer look into the subject of risk perception in terms of flood risk management but I would also like to suggest two alternative ways for classification:

Either you go back to your first definition with high, average and low 'exceedance probability' which you anyway still use to explain the scenarios and name the maps in section 3.2 or you define the multi-model combination scenarios based on 'severity' (I = low severity, II = average severity, III = high severity). In both ways it can still be discussed that risk perception of the decision maker will then influence the choice of mitigation measures. I still find it problematic that the occurrence of the scenario with a high exceedance probability is – according to my interpretation – theoretically less probable than the occurrence of the scenario with low exceedance probability. Or in other words: just looking at the used M%-outputs in each scenario it's more likely that the real scenario will exceed scenario I than to "fall below" scenario III. Therefore, although uncertainties of the forecast are considered, the method as presented rather provides underestimating hazard maps as rather low ensemble members (M%-hydrographs) are used. In general, the same is true if you classify by 'severity' but here you don't explicitly make a statement about the probability. The table below should exemplify what to my opinion would be a "balanced" selection of M%-scenarios representing uncertainties in a more proper way. But I can imagine that changing this would maybe not be manageable and therefore I would suggest to use 'severity' for classification and maybe discuss the mentioned issue concerning the used probabilities.

| Scenario | severity /exc. Prob | Building class | | | |
|---|---|---|---|---|---|
| | | I | II | III | IV |
| I | low / high | 10% | 30% | 50% | 70% |
| II | average | 20% | 40% | 60% | 80% |
| III | high / low | 30% | 50% | 70% | 90% |

I would also suggest to again proof read the paper and check the use of the correct numbers (see e.g. p. 6, line 15 ->"75%, 90%, and 95%").

References:

Botzen, W. J. W., Aerts, J. C. J. H., & van den Bergh, J. C. J. M. (2009). Dependence of flood risk perceptions on socioeconomic and objective risk factors. *Water Resources Research*, *45*(10), 113. https://doi.org/10.1029/2009WR007743

---

## Author Response (AR2)

Authors Response

We appreciate the constructive feedback from the reviewer in improving the quality of the manuscript. We have addressed the review comments in the revised manuscript.

Reviewer 1

5   I would like to thank the authors for giving answer to all my concerns and for implementing many of the suggested corrections. In this new version, it is much clearer with which hydrological input the hydraulic models were run and there is now much less confusion about this part.

However, in his comments to the resubmission, the editor doubted the suitability of 'risk perception' used as replacement for 'exceedance probabilty'. I share his doubts. The paper cited with context to risk perception (Paulsen et al., 2012) is maybe not

10   the best choice, as this is about "the development of economic risk preference from childhood to adulthood" with a very economic focus and I would say that in flood risk management stakeholders or decision-makers would maybe not select a map that is described as 'risk-seeking' to initiate mitigation measures. From the view of a decision-maker, I should not have to define if I am risk-seeking, -neutral or –averse to select a suitable hazard map for initiating mitigation measures, instead I should receive maps that show what can potentially happen in an objective way. Of course the selected strategy then depends

15   on the individual risk perception. There are other, to my opinion more suitable papers about risk perception with the context to flood risk management (see e.g. Botzen et al 2009). The authors should have a closer look into the subject of risk perception in terms of flood risk management but I would also like to suggest two alternative ways for classification:

Either

1. you go back to your first definition with high, average and low 'exceedance probability' which you anyway still use
20      to explain the scenarios and name the maps in section 3.2 or
2. you define the multi-model combination scenarios based on 'severity' (I = low severity, II = average severity, III = high severity).

**In both ways it can still be discussed that risk perception of the decision maker will then influence the choice of mitigation measures.**

25   I still find it problematic that the occurrence of the scenario with a high exceedance probability is – according to my interpretation – theoretically less probable than the occurrence of the scenario with low exceedance probability. Or in other words: just looking at the used M%-outputs in each scenario it's more likely that the real scenario will exceed scenario I than to "fall below" scenario III. Therefore, although uncertainties of the forecast are considered, the method as presented rather provides underestimating hazard maps as rather low ensemble members (M%-hydrographs) are used.

30   In general, the same is true if you classify by 'severity' but here you don't explicitly make a statement about the probability. The table below should exemplify what to my opinion would be a "balanced" selection of M%-scenarios representing uncertainties in a more proper way. But I can imagine that changing this would maybe not be manageable and **therefore I would suggest to use 'severity' for classification and maybe discuss the mentioned issue concerning the used probabilities**.

| Scenario | severity /exc. Prob | Building class | | | |
|---|---|---|---|---|---|
| | | I | II | III | IV |
| I | low / high | 10% | 30% | 50% | 70% |
| II | average | 20% | 40% | 60% | 80% |
| III | high / low | 30% | 50% | 70% | 90% |

Authors: Thank you for your suggestion, we have chosen your first suggestion to use high, average and low 'exceedance probability' and have used the Botzen et al. 2009 for risk perception. In section 3.2

"Each scenario presents a given risk perception that can be defined as the subjective judgement of a decision-maker about the severity of the risk, which can influence the choice of mitigation measures (Botzen and van den Bergh, 2009)."

We agree that there can be numerous combination and added in conclusion that a pre-assessment of combination scenarios should be carried out. In conclusion

"In order to design the multi-model combination, a group consisting of researchers, operational bodies and experts in the field of flood risk management should be consulted"

I would also suggest to again proof read the paper and check the use of the correct numbers (see e.g. p. 6, line 15 ->"75%, 90%, and 95%").

Authors: Thank for pointing out error, we have corrected it and proof-read the updated manuscript.

References:

[revised manuscript text omitted]